computational biology/computer modelling and simulation

COVID-19, contact tracing, test-trace-isolate, non-pharmaceutical interventions

**Author for correspondence:**
Yee Whye Teh
e-mail: y.w.teh@stats.ox.ac.uk

†First authors, equal contribution, order randomized.

# Effectiveness and resource requirements of test, trace and isolate strategies for COVID in the UK

Bobby He[1,†], Sheheryar Zaidi[1,†], Bryn Elesedy[2,†], Michael Hutchinson[1,†], Andrei Paleyes[3,†], Guy Harling[4], Anne M. Johnson[4], Yee Whye Teh[1] and on behalf of the Royal Society's DELVE group[1]

[1]Department of Statistics, and [2]Department of Computer Science, University of Oxford, Oxford, UK
[3]Department of Computer Science and Technology, University of Cambridge, Cambridge, UK
[4]Institute for Global Health, UCL, London, UK

AP, 0000-0002-3703-8163; YWT, 0000-0001-5365-6933

We use an individual-level transmission and contact simulation model to explore the effectiveness and resource requirements of various test-trace-isolate (TTI) strategies for reducing the spread of SARS-CoV-2 in the UK, in the context of different scenarios with varying levels of stringency of non-pharmaceutical interventions. Based on modelling results, we show that self-isolation of symptomatic individuals and quarantine of their household contacts has a substantial impact on the number of new infections generated by each primary case. We further show that adding contact tracing of non-household contacts of confirmed cases to this broader package of interventions reduces the number of new infections otherwise generated by 5–15%. We also explore impact of key factors, such as tracing application adoption and testing delay, on overall effectiveness of TTI.

## 1. Introduction

The 2019 COVID pandemic presents an unprecedented challenge to societies and governments worldwide. COVID is an infectious disease caused by severe acute respiratory syndrome coronavirus 2 (SARS-CoV-2). Without a widespread vaccination roll-out nor effective therapeutics, as was the case in 2020, non-pharmaceutical interventions (NPIs) are the essential tools by which a policy maker can combat the transmission of this highly infectious virus.

**Table 1.** Main decision points defining the TTI strategies considered.

| TTI strategy | no TTI | symptom-based TTI | test-based TTI | test-based TTI test contacts |
|---|---|---|---|---|
| Isolate individual on symptoms? | yes | yes | yes | yes |
| Quarantine household on symptoms? | yes | yes | yes | yes |
| Test symptomatic individuals? | no | yes | yes | yes |
| Trace contacts on symptoms? | no | yes | no | no |
| Trace contacts on positive test? | no | no | yes | yes |
| Quarantine traced contacts? | no | yes | yes | yes |
| Test contacts? | no | no | no | yes |

NPIs have been aimed at reducing social contact, with examples including stay-at-home orders, working from home, travel restrictions, and closing of non-essential businesses, schools, universities and workplaces. While they hold the potential to reduce transmission [1,2], the amount by which they reduce transmission and the social and economic costs they incur are yet to be fully understood.

Test-trace-isolate (TTI) systems are an interconnected range of NPIs that aim to prevent community transmission and reduce the effective reproduction number. TTI systems work by identifying infected individuals (test), identifying their social contacts (trace), and isolating the infected individuals and quarantining their contacts (isolate) to prevent onward transmissions. TTI systems have proven to be an effective component of an overall NPI strategy, particularly when community transmission is low [3–5] and have emerged as a key lever for policy makers in 2020. However, the efficacy, resource requirements of TTI systems as well as their various impacts on society vary significantly with their implementation and the size of the epidemic, and in this aspect TTI is understudied.

In this study, we aim to quantify how the effectiveness and resource requirements of TTI systems vary with respect to implementation and in conjunction with other NPIs. We focus on TTI in the context of high infection numbers and significant community transmission, such as in the UK in May 2020. In particular, we study the resource requirements of TTI systems including the number of tests needed, the required number of contacts traced and the number of person-days spent under quarantine, and the corresponding reductions in transmissions.

We use a simulation-based model that builds upon the individual-level model of [6] and stratifies transmissions by setting (household, work, school, other) using the BBC Pandemic data of 40 162 participants in the UK. We account for recent research on the timeline of COVID infections as well as various logistical and temporal aspects of real-world implementations of TTI strategies. These include baseline symptom presentation in the COVID-free population, for instance due to a common cold; non-uniform infection profiles over time; imperfect compliance with symptom reporting, isolating and quarantining; and delays associated with reporting symptoms, testing and tracing. We consider three potential TTI strategies, detailed in table 1, which differ in how they trade-off the speed of tracing contacts against the resource requirements associated with TTI.

Under optimistic but plausible assumptions, our analysis suggests that TTI has a moderate impact on reducing transmission, with the majority of reductions being due to the isolation of infected individuals along with quarantining of their household contacts. We find that TTI can be an important component of an overall strategy to combat the spread of COVID, particularly if the reproduction number is around 1, but that TTI alone is not sufficient to contain the epidemic. Moreover, our sensitivity analysis demonstrates that the effectiveness of a TTI strategy is dependent on two key factors: the speed of the system, specifically that of testing and manual tracing, and public compliance and engagement with the system.

## 2. Related works

Various previous studies have examined the effectiveness of TTI as a strategy to contain COVID. Initial work, such as [7], focused on TTI efforts to contain the risk arising from imported cases and so do not consider TTI in tandem with other NPIs. Similarly, Ferretti *et al*. [8] did not model the prospect of non-TTI NPIs and studied the use of digital app tracing to contain a COVID epidemic. McLachlan *et al*. [9] propose using Bayesian networks as a modelling tool to combat the 'incomplete information' problem, but do not provide a detailed analysis of cost or effectiveness. Recent research has also

studied the spread of COVID in specific countries. For example, Ribeiro *et al.* [10] study the effect of city size on the spreading dynamics of COVID in Brazil, and Hâncean *et al.* [11] profile early cases in Romania, concluding that migration between Romania and Italy was important for COVID transmission.

More relevant to this current work, Kucharski *et al.* [6] provides a framework (which we build on) using real-world primary-secondary contact pairs data in order to study the impact of combining TTI with other NPIs, but did not consider the temporal and logistical considerations of a practical implementation of TTI. Elsewhere, similar conclusions about the overall effectiveness of TTI for reducing $\mathcal{R}$ where made by backpropagation model from the UK's Department of Health & Social Care [12].

Considerations such as changing infectiousness levels during the infectious period and the delay in receiving test results are modelled in [13], but they are unable to model realistic NPIs, such as working from home, due to a more basic contact generation procedure. In a follow-up work concurrent to this present study, Kretzschmar *et al.* [14] analyse the impact that logistical delays in the TTI procedure have on effectiveness in terms of reducing $\mathcal{R}$, and similarly conclude that it is crucial to minimize any such delays in order to maximize the effectiveness of TTI. Kretzschmar *et al.* [14] did not study the resource requirements of different TTI strategies nor the possibility of symptomatic COVID negative primary cases entering any given TTI system.

Disease control measures, including TTI, were a subject of active research before COVID. A large body of literature was created in the aftermath of the SARS epidemic in 2003. Eames & Keeling [15] suggested an analytical model of effectiveness of contact tracing on infectious disease control, looking at networks of various types: homogeneous and mixed (by gender), with and without clusters. Fraser *et al.* [16] analyse effects of a range of intervention scenarios. Armbruster & Brandeau [17] frame deployment of contact tracing as a cost optimization problem. Unfortunately applicability of these works to the situation around COVID is limited, because none of them take mobile phone applications as a tracing tool into account. This is understandable considering the penetration and capabilities of mobile technology in years 2003–2007 compared with 2020.

# 3. Methods

Our simulation model consists of three stages: generation of the characteristics of primary cases, generation of the contacts of the primary cases and the application of test-trace-isolate strategies to the primary cases and their contacts. We specialize the setting of our model to what might be expected during summer months (June–August) in the UK, and to five scenarios corresponding to different levels of stringency of other NPIs.

## 3.1. Generation of primary cases

We assume a total of 20k new COVID infections each day, split between symptomatic and asymptomatic cases.[1] This number is around the upper bound estimated by Flaxman *et al.* [1] for 4 May. As there is no consensus for proportion of asymptomatic COVID cases, we followed [6] and set this at 40%, with asymptomatic infectiousness reduced by 50% relative to symptomatic cases.

Alongside new COVID positive cases, we include a baseline of 100k COVID negative primary cases who present COVID-like symptoms and may enter a given TTI system, thereby increasing resource requirements. 100k is around the estimated pre-pandemic number of individuals presenting symptoms of fever or cough on any given day over the summer period in the UK according to Bug Watch [18]. While current COVID NPIs are believed to have reduced the presentation of other respiratory illnesses, as demonstrated in Hong Kong by Cowling *et al.* [19], 100k is a reasonable worst case scenario for summer. For the infection timeline of each COVID positive primary case, we assume a latent period of 3 days, a mean duration of 2 days of presymptomatic infectious period before reporting symptoms, and a non-uniform infection profile over 10 days peaking on the day before the expected day of symptom presentation [20].

## 3.2. Generation of contacts

We followed the model of [6] for our contact generation. In summary, we use the BBC Pandemic dataset [21], which contains data on the social contacts of 40 162 UK participants, to simulate the number of daily

---

[1]Throughout the paper, we use the term *case* to refer to a generic individual considered within our model, whether COVID positive or not. Whereas, *infection* refers to a COVID positive individual that is not necessarily confirmed to be positive.

close contacts of the primary case. The total number of daily contacts for the primary case is broken down into the following categories: household, work/school and other. To simulate secondary cases, we assume that each contact of the primary case has a probability, known as the secondary attack rate (SAR), of being infected over the course of the infectious period, independent of the remaining contacts. We assume new non-household contacts for each day of the simulation, whereas household contacts are assumed to have repeated contact with the primary case on each day of simulation. As in [6], we have separate SARs for household and non-household contacts to model the fact that the close and repeated nature of household interaction implies household contacts should be at greater risk of secondary infection. We set our SARs to give a base $\mathcal{R}$ of 3.87 in a no NPIs scenario following [1], while maintaining the same proportion (15%) of total infections that are intra-household as in [6]. This leads to a doubling time of 3.35 days in a no NPIs scenario.

Detailed descriptions of the generation of primary cases and their contacts are given in appendix A.

## 3.3. Scenarios of other non-pharmaceutical interventions

We consider five scenarios, each with a different combination of NPIs, corresponding to varying levels of stringency. These scenarios consider different levels of guidance for physical distancing, working from home and school closures, and range from the most stringent (S5) which models the lockdown scenario prior to 9 May, to medium stringency (S3) which models a scenario with more social contacts, 50% of schools being open and 45% of the working population working from home, and S1 which models no NPIs except for households being quarantined at home on presentation of symptoms. Detailed description of the scenarios are given in appendix B.

## 3.4. Test-trace-isolate strategies

The three core TTI strategies that we will analyse are summarized as follows:

— Symptom-based TTI: Start contact tracing and quarantine contacts as soon as a primary case reports COVID-like symptoms.
— Test-based TTI: Start contact tracing and quarantine contacts once a primary case is confirmed by a test to be COVID positive.
— Test-based TTI with contact testing: Start contact tracing and quarantine contacts once primary case is confirmed by a test to be COVID positive. Test the contacts of a confirmed COVID positive primary case.

For each strategy, including no TTI, the primary case and members of their household are asked to isolate/quarantine at home when the primary case first presents symptoms, following current UK government guidelines (as of 17 May 2020). Contact tracing commences at either symptom presentation or test returning positive, and all traced contacts are asked to quarantine for a total of 14 days. If traced contacts show symptoms they are entered into the TTI system as primary cases themselves. If contacts are tested and they test negative, they are released from quarantine. We assume isolation and quarantines prevent all subsequent transmissions.

We model both NHSx app-based and manual contact tracing, assuming that 35% of the population will download and regularly use the app. This is estimated assuming that around 60% of the population downloads the app (this is around the proportion of the Isle of Wight population who have downloaded the NHSx app on 10 May), and 60% of those downloads are regularly using it (this is on the lower side of estimates of usage of the Zoe app). We assume a compliance level of 80% for both symptom reporting as well as requests to quarantine or isolate. The success of TTI is highly sensitive to the level of public compliance, as demonstrated in figures 4 (Top Right) and 10 (Right), which can in turn be affected by appropriate public messaging, incentives and coordination with employers. Note that a concurrent study [22] found lower UK public compliance levels than 80% in practice between March and August 2020, which complements our findings that maximizing public compliance is a key vehicle to improving the effectiveness of TTI strategies. Finally, we assume that the time taken to obtain a test result is 2 days, and that it takes 1 day following this for contacts to be manually traced (app tracing is assumed instantaneous).

Diagrams of these strategies can be found in figure 1, and a table describing the options can be found in table 1. Details along with larger versions of the diagrams are given in appendix C.

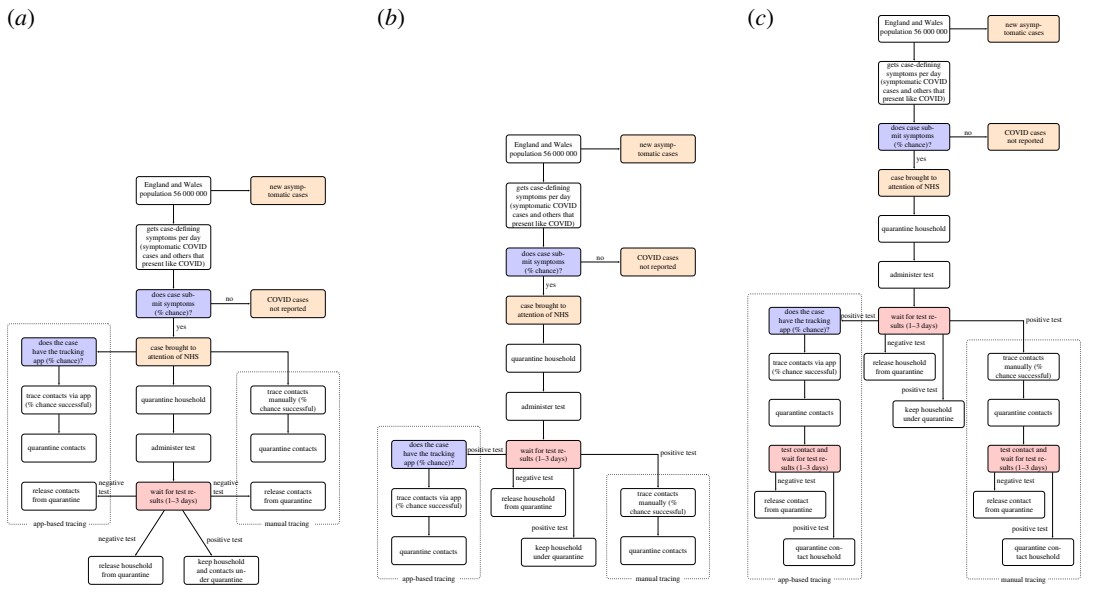

**Figure 1.** Diagrams detailing the flow of new cases, their household contacts and non-household contacts through the various TTI system. (*a*) Diagram of an individual passing through a symptom-based TTI system. (*b*) Diagram of an individual passing through a test-based TTI system. (*c*) Diagram of an individual passing through a test-based TTI system with contact testing.

## 3.5. Limitations

There are several limitations to our simulation study, both in terms of simulating transmission dynamics and assumptions made regarding different TTI strategies:

— Our model only simulates a single generation of transmission, and does not model subsequent infections of tertiary cases, nor the effects that complex social networks will have on the spread of COVID in society.
— As in [6] we assume non-household contacts are only met once during the infectious period of the primary case. This will impact both the number of contacts needed to be traced in a TTI strategy and the timeline of infection because repeated contacts, like household contacts, are likely to be infected earlier in a primary case's infectious period.
— Our simulations suppose that once a primary case is isolated all potential future infections are prevented. This may be unrealistic, especially for household contacts. Note that our model includes a chance that the case may not comply with an advice to self-isolate.
— Our model assumes that an individual's ability to work from home is independent of their number of daily work contacts.
— There is much that is not yet fully understood surrounding the dynamics of COVID transmission, including: the proportion of COVID positive cases that are asymptomatic, the infectiousness of asymptomatic cases and the infectiousness profile of a COVID positive individual.
— Our model assumes the beginning of a primary case's infectious period is known when they report symptoms. This is unlikely to hold in practice, but may not be a bad approximation under the assumption of a 2-day presymptomatic infectious period. This has implications for the total number of contacts that need to be traced, as well as the number of person-days spent in quarantine.
— It is likely that lockdown and social distancing measures have led to significant decreases in other respiratory illnesses, as suggested by Cowling *et al.* [19]. However, we cannot accurately predict the likely impact of COVID NPIs on the prevalence of COVID-like symptoms in the general population over the next year, and our model was based on typical levels in previous years. This has implications for the number of tests required, as well as manual tracers and quarantine days required of symptom-based TTI strategies.
— We do not account for the varying prevalence of COVID across different regions, demographics and sectors, as well as the varying risk factors of COVID for different individuals. A surveillance system can be important, both in the identification and management of local outbreaks, and in the

**Table 2.** Comparison of effective $\mathcal{R}$ for different TTI strategies, for scenarios S5 (most stringent NPIs in place) to S1 (no NPIs). Shown are mean $\mathcal{R}$ along with 95% confidence intervals over our 20k primary cases, $\mathcal{R} < 1$ values are highlighted in bold. It can be clearly seen that control of the virus spread is only possible if TTI is used in conjunction with other NPIs.

| TTI strategy | S5 | S4 | S3 | S2 | S1 |
|---|---|---|---|---|---|
| no TTI | **0.78 ± 0.03** | 1.14 ± 0.04 | 1.59 ± 0.04 | 2.01 ± 0.05 | 2.34 ± 0.06 |
| symptom-based TTI | **0.66 ± 0.02** | **0.94 ± 0.03** | 1.26 ± 0.04 | 1.65 ± 0.04 | 1.94 ± 0.05 |
| test-based TTI | **0.69 ± 0.03** | **0.98 ± 0.03** | 1.37 ± 0.04 | 1.73 ± 0.04 | 2.02 ± 0.05 |
| test-based TTI, contact testing | **0.69 ± 0.03** | **0.99 ± 0.03** | 1.35 ± 0.04 | 1.76 ± 0.05 | 2.04 ± 0.05 |

incorporation of a spatio-temporal predictive model for $\mathbb{P}(\text{COVID positive} \mid \text{symptoms and covariates})$ to help improve the efficiency of a resource-constrained TTI system.

— It is difficult to gauge public compliance towards a given TTI strategy. There are various socio-economic factors that may need to be considered here, such as if contacts advised to isolate will be compensated for lost income while they are quarantined. A recent study by Bodas & Peleg [23] suggested that public compliance towards self-isolation in Israel would drop from 94 to 57% if compensation was removed. Moreover, a concurrent study [22] suggests our baseline compliance levels, on which we perform sensitivity analysis in appendix E, are optimistic compared with observed levels in the UK in the summer months of 2020.

— Throughout our analysis, we assume a compliance rate that is the same for all parts of the test and trace system, i.e. we assume a person either complies with the full suite of NPI measures, or does not. It is not clear, however, that this is the case; for example, Smith *et al.* [22] provides significant evidence that the propensity to provide contact details is much higher than the propensity to self-isolate.

## 4. Results

We first compare our three proposed TTI strategies (with default parameter settings) against no TTI, which is the current (as of 17 May 2020) UK guideline of isolating primary cases and quarantining household contacts when the primary case becomes symptomatic but no contact tracing. Table 2 shows the effective $\mathcal{R}$ across the five stringency levels S5 to S1. As we can see, the addition of contact tracing and quarantining leads to modest reductions in $\mathcal{R}$ across all five scenarios, and TTI should only be deployed as part of a wider package of NPIs to keep $\mathcal{R}$ below 1 and control the epidemic. As an aside, even modest reductions in $\mathcal{R}$ can lead to substantial reductions of absolute infection numbers over time (see appendix G for the expected number of new primary cases across time for different values of fixed $\mathcal{R}$ during the exponential growth of the epidemic).

Figure 2 compares effective $\mathcal{R}$, number of contacts manually traced, number of tests needed and number of person-days that contacts spent in quarantine. In terms of resource requirements, it is clear that symptom-based TTI requires significantly more manual traces and person-days in quarantine compared with test-based TTI. This is due to the non-specific nature of COVID symptoms. The low specificity also has implications for compliance with TTI guidance, and we expect lower compliance with guidance in symptom-based TTI, which can in turn have a larger impact on $\mathcal{R}$. When considering test-based TTI with and without contact testing, a trade-off emerges between the extra tests needed and the gained ability to safely release from quarantine the uninfected contacts of COVID positive cases. However, antigen testing is insensitive during the incubation period, so repeat testing will be necessary to avoid infected contacts being released prematurely [24].

To understand the contribution of various measures and parts of the TTI system to the overall transmission rate, we also consider a decomposition of the total number of *potential* transmissions that could have occurred into six categories. Note that each potential transmission is either prevented or occurs (given a set of measures). If prevented, the reason is one of the following three: social distancing NPIs (including working from home and reductions in non-household contacts), isolation of symptomatic index case and quarantining of their household, or secondary extra-household contact tracing and quarantining. If the transmission occurs, the reason is one of the following three: index case being asymptomatic, index case being symptomatic but failing to report symptoms, or transmission arising due to imperfect contact tracing for a symptomatic index case. More specifically,

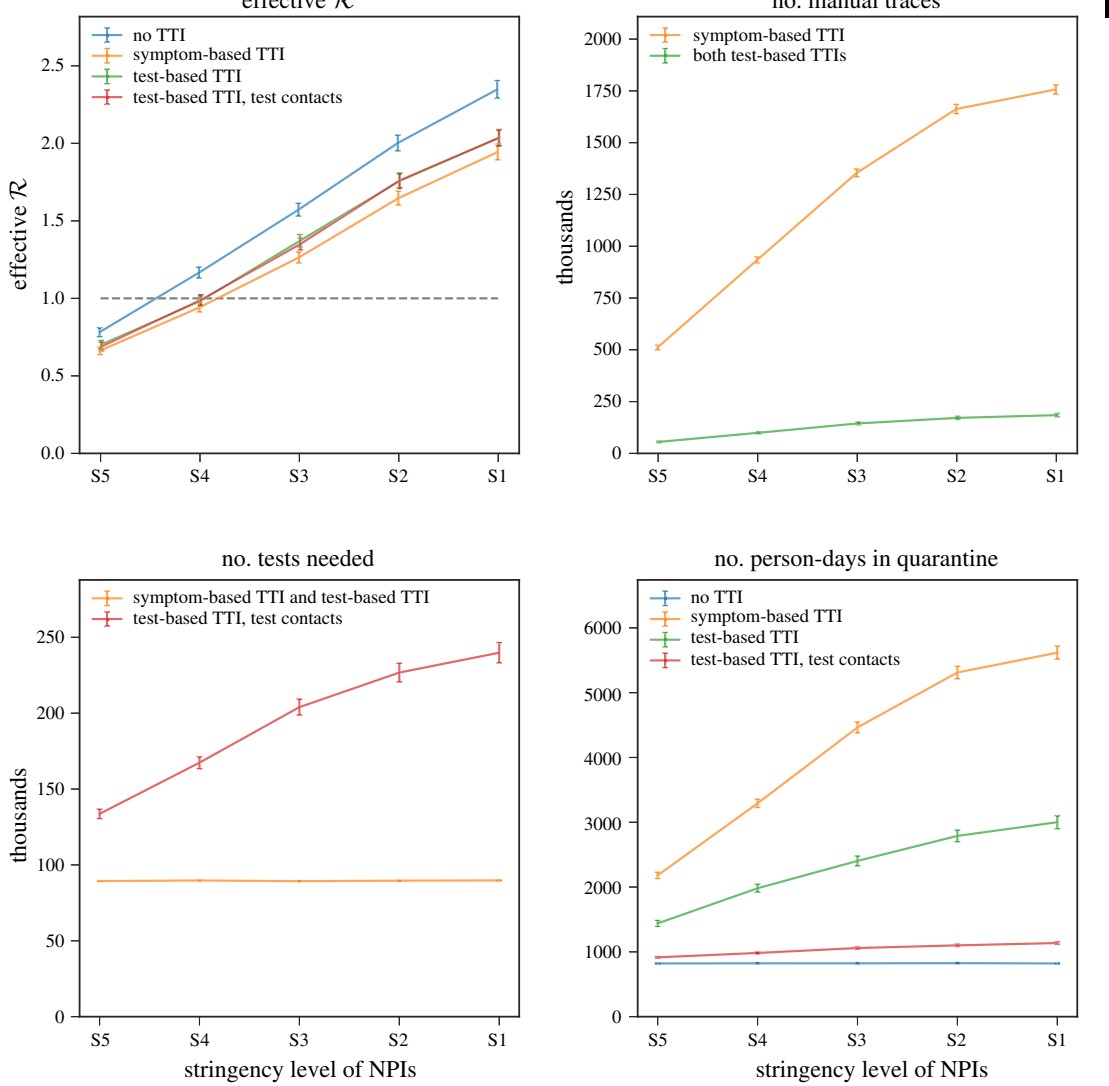

**Figure 2.** Impact on effective reproduction number $\mathcal{R}$ and resource requirements of various TTI strategies, across five sets of NPIs with different stringency levels, for 100k primary cases with symptoms but COVID negative and 20k COVID positive primary cases, as described in §A.1. Resource requirements are displayed in thousands.

imperfect tracing is due to secondary cases that are never traced, delays in tracing and traced secondary cases being non-compliant.

Figure 3 shows this breakdown for each of the five NPI stringency levels assuming test-based TTI is in place. As shown, social distancing is responsible for a large portion of prevented transmissions for stringency levels S3–S5. As the stringency level of the NPI is reduced to S1–S2, index case isolation alongside quarantining of their household becomes responsible for the majority of prevented transmissions. Note we have separated the effects of base social distancing (reducing social contacts, closure of shops, etc.) from isolating index cases to demonstrate the scenario in which regular social distancing is not used, but index cases are still isolated, e.g. S5. The reversed scenario is unlikely to ever be used. The combination of these two is what would colloquially be referred to as 'social distancing'. For all stringency levels, tracing is responsible for a relatively small portion of the overall prevented transmissions. On the other hand, of transmissions that do occur, the majority (around half) are due to asymptomatic index cases across all stringency levels (this depends crucially in the model on the assumption of 40% of COVID positive cases being asymptomatic with halved infectiousness), followed by transmissions from symptomatic index cases that failed to report (which depends on compliance levels within the population).

Having demonstrated that TTI should be adopted in unison with other NPIs, and that test-based TTI is a good compromise between resource requirements and reductions in $\mathcal{R}$, we now analyse three specific

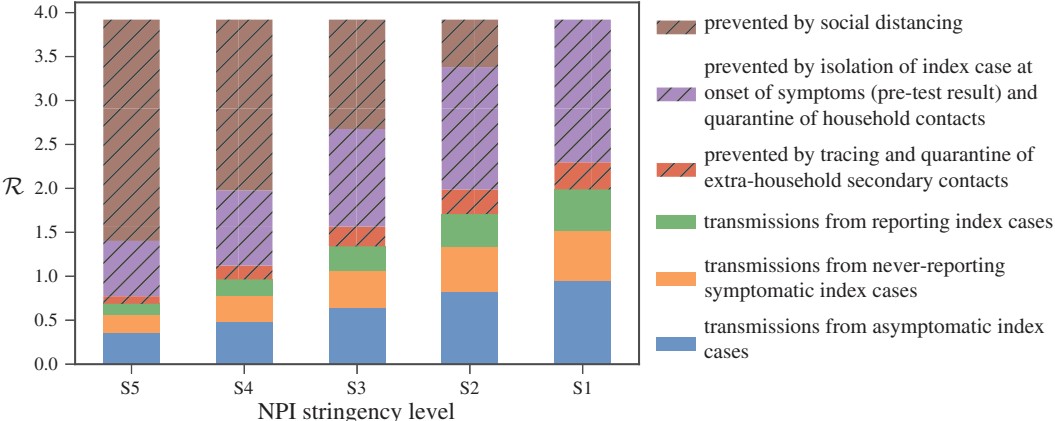

**Figure 3.** Analysis of how potential cases are prevented or transmitted and by what means for the five stringency levels of lockdown with test-based tracing. Proportions of transmissions are represented in terms of their contribution to the effective $\mathcal{R}$. Hatched bars indicate infections *prevented* by each stringency level.

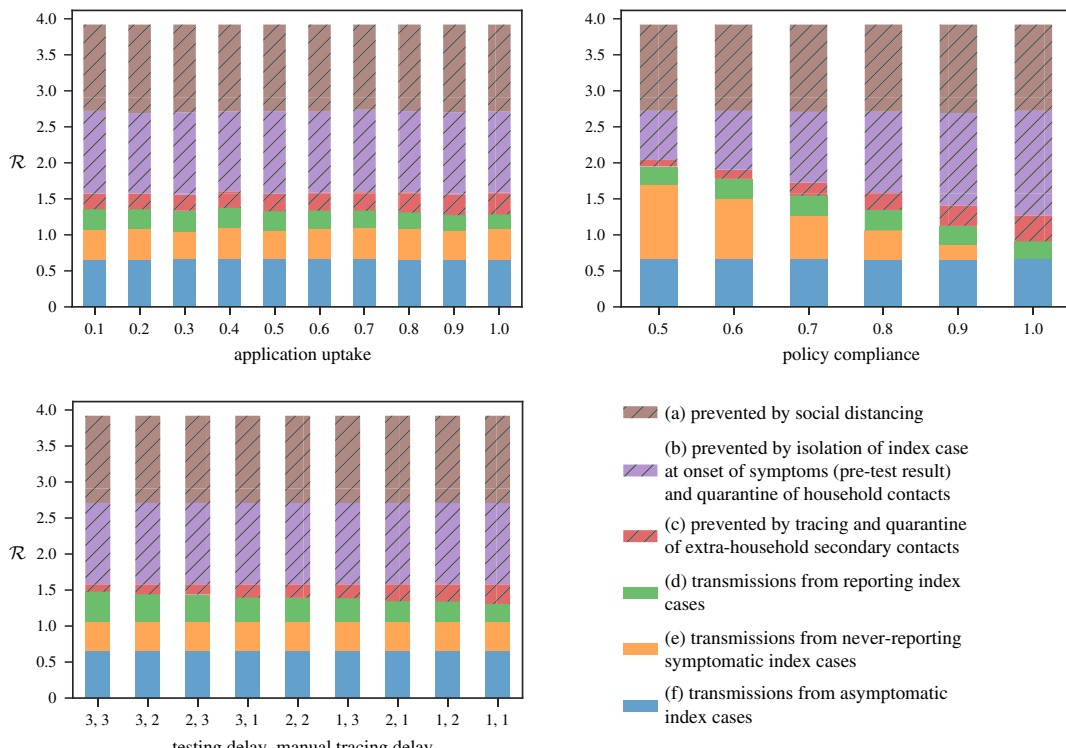

**Figure 4.** The impact on the percentage of ongoing transmission reduced by TTI of changing the application uptake rate, the policy compliance rate, and reducing delays for testing and manual tracing for the S3 severity levels and using the test-based TTI strategy.

areas in which policy can help to improve the effectiveness of test-based TTI: the time delay in testing and tracing, the level of public compliance, and the uptake of the app. Figure 4 shows the contributions to $\mathcal{R}$ from each NPI and part of the TTI system for the S3 stringency scenario, while figure 5 shows the resulting $\mathcal{R}$ across the five scenarios as we vary test/trace delays and compliance.

We find that the most important factor determining TTI effectiveness is the level of public compliance with TTI guidance to report symptoms, get tested, isolate and quarantine. Figures 5 (right) and 4 (top right) highlight the benefit of increasing public compliance towards TTI measures. There is a clear reduction in $\mathcal{R}$ across all TTI strategies as compliance is increased in the S3 scenario.

We next evaluate the impact on $\mathcal{R}$ of the time delay in testing and manual tracing for the test-based TTI strategy. The results are shown in figures 5 (left) and 4 (bottom left). The testing delay is the time between the primary case reporting symptoms and the results of a test being returned, while the

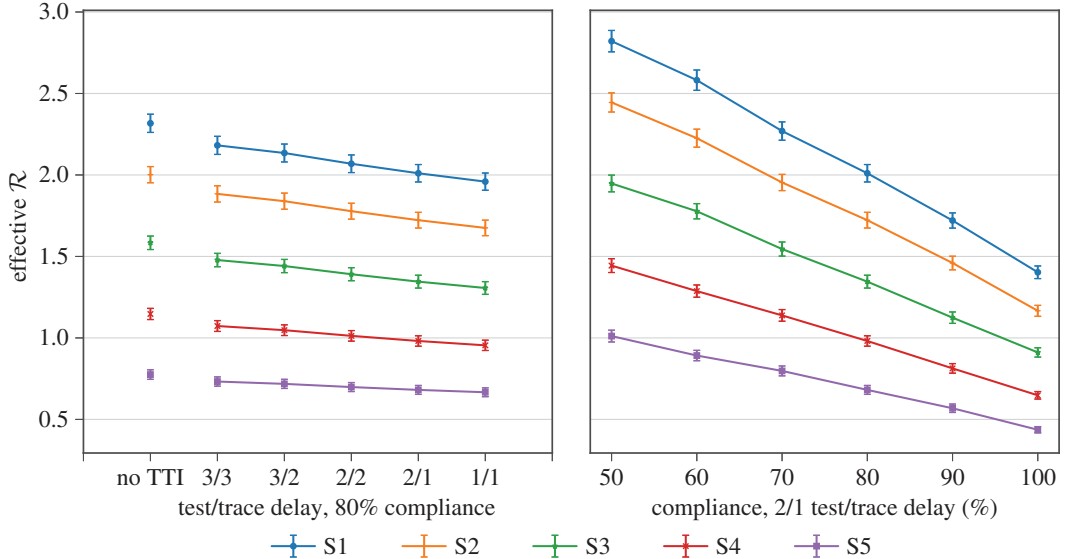

**Figure 5.** Impact of testing/tracing delays and compliance on $\mathcal{R}$, for the test-based TTI strategy.

manual tracing delay is the time between a primary case being confirmed COVID positive and the identification and quarantining of their contacts. The results indicate that, in order for the test-based TTI strategy to be effective, both of these delays should be reduced. While the impact of TTI in proportion to social distancing and primary case isolation is small, its effect in bringing down the remaining transmission not caught by these measures can be significant. In particular, for example, the addition of an effective TTI system in stringency level S2 can make the difference between $\mathcal{R} > 1$ and $\mathcal{R} < 1$ and therefore the difference between containing the pandemic and not. In the context of less stringent NPIs such as S1–S2, TTI can be particularly important as social contacts are more numerous and containment relies heavily on testing and tracing. We find non-negligible reductions in the effective $\mathcal{R}$ across all NPIs considered when the delays are reduced to a total of 2 days turnaround.

Figure 4 (top left), shows the effect on $\mathcal{R}$ of changes in app uptake. We see that there is a slight downward trend in mean effective $\mathcal{R}$ as app uptake is increased, keeping all other parameters constant. One reason for this is that our default setting for manual tracing delay is just 1 day, compared with no delay for app-based tracing. If viewed in the context of a longer manual trace delay, this chart would further emphasize the critical role of the app in reducing $\mathcal{R}$. We note that under the current system, an increase in app usage does not cause a reduction in the number of manual traces required. The primary case will not know which of their contacts have been traced through the app, nor will the manual tracers, therefore it is still necessary to trace manually as many contacts as possible. On the other hand, the app will help trace those contacts that are unable to be manually traced, e.g. those unknown to the primary case. Better coordination between app-based and manual tracing systems could potentially reduce the resulting manual tracing effort required.

Note that in figure 4 the overall effects on $\mathcal{R}$ of test/trace delays and app uptake are relatively small compared with the effect of policy compliance. This is because, while the effects of both on the subpopulation that is known to the TTI system are substantial, this subpopulation represents a relatively small part of the entire population. Other details on the effects of compliance, delays and app uptake are presented in appendix E.

## 5. Conclusion

In this work, we modelled the effects of various TTI strategies in combination with other NPIs in order to assess their effectiveness and resource requirements.

On the effectiveness of TTI, we observe that across the range of scenarios considered, TTI has a moderate effect on $\mathcal{R}$, and implementation along with other NPIs will be necessary to control the COVID epidemic in the UK. Implemented on top of current UK government recommendations to self-isolate and quarantine households on COVID symptoms, test-based TTI strategies reduce $\mathcal{R}$ between 10 and 15%, while symptom-based TTI reduces between 15 and 20% (table 2). For example, in

medium stringency scenario S3, symptom-based TTI reduces $\mathcal{R}$ from $1.59 \pm 0.04$ to $1.26 \pm 0.04$ and test-based TTI to $1.37 \pm 0.04$. The most significant reduction in transmissions of a TTI system is due to prompt self-isolation of a symptomatic case and the quarantining of their household.

Our analysis shows that two main factors determine the effectiveness of TTI strategies:

— The amount of time required for testing and for manual contact tracing plays a significant role in the effectiveness of TTI (figure 5, left). A reduction of time between symptom onset and contact being informed to isolate from 5 to 3 days leads to a 60–70% improvement in effectiveness of a test-based TTI strategy in our simulation. For example, in scenario S3 a 5-day delay has an effective $\mathcal{R}$ of $1.46 \pm 0.04$ while a 3-day delay has $1.37 \pm 0.04$, relative to $1.59 \pm 0.04$ with no TTI.
— TTI performance is strongly dictated by its coverage of transmission chains and compliance of the general population with its guidance (figures 5, right and 4, top right). Leakages from the system include asymptomatic COVID positive cases, symptomatic cases who do not report symptoms and imperfect contact tracing (e.g. of contacts unknown to the primary case). To maximize effectiveness of the TTI system, it is crucial to maximize app uptake and compliance to reduce leakages from the system.

We also looked at resource requirements of TTI systems. To that end we show that if uptake of a contact-tracing app is insufficient, manual contact tracing is necessary and is the main resource requirement of TTI strategies. A typical baseline of COVID-like symptoms among the general COVID negative population means that symptom-based TTI has low specificity and requires significantly higher numbers of manual contact tracings and person-days quarantined (figure 2). Further, in a test-based TTI strategy, additional testing contacts has a marginal impact on $\mathcal{R}$ in our simulation (due to identification of asymptomatic COVID positive contacts) but can significantly reduce the number of person-days of contacts quarantined (figure 2). Testing too early in the incubation period, and likely variability in the length of incubation periods [24], might, however, lead to missing infected contacts, and repeat testing is required.

We perform sensitivity analyses where appropriate in appendix F, but our key findings above are not substantively affected.

Data accessibility. Data and relevant code for this research work are stored in GitHub: https://github.com/rs-delve/tti-explorer and have been archived within the Zenodo repository: https://doi.org/10.5281/zenodo.4560214.

Authors' contributions. B.H., S.Z., B.E., M.H., A.P. and Y.W.T. implemented the model and ran the experiments. G.H. and A.M.J. conceptualized the model flow and provided policy advice throughout.

Competing interests. The authors declare no competing interests.

Funding. B.H. is supported by the EPSRC and MRC through the OxWaSP CDT programme (EP/L016710/1). S.Z. is supported by Aker Scholarship. B.E. is supported by the UK EPSRC CDT in Autonomous Intelligent Machines and Systems (grant reference EP/L015897/1). M.H. is supported by the ESPRC and Qualcomm through the StatML CDT programme (EP/S023151/1). A.P. is supported by Alan Turing Institute and DeepMind (via University of Cambridge Endowment Fund). G.H. is supported by a Sir Henry Dale fellowship from the Royal Society and the Wellcome Trust (210479/Z/18/Z).

Acknowledgements. This work was done as a part of the 'Test, Trace, Isolate' report from the Royal Society DELVE group. We are grateful to the members of said group for comments and discussions.

Disclaimer. We have read the information for authors and publish policy. We all agree with the policy and prepare the manuscript in accordance with the guidance.

# Appendix A. Case and social contact generation

We use a single-stage transmission model consisting of a simulated primary case with simulated social contacts. We assume a primary case that is COVID positive is infectious for 10 days, with our simulation starting when the primary case becomes infectious and ending 10 days thereafter.

For each primary case, we simulate whether they are asymptomatic COVID positive, symptomatic COVID negative, or symptomatic COVID positive. We also simulate their age and whether they will report COVID-like symptoms (should they have them) during the simulation. If the primary case does report symptoms, we further simulated the day on which they report them. Social contacts made during the infectious period are generated conditional on the age of the case and are categorized as home, work/school and other. The number of daily contacts made in each of these categories is sampled according to the BBC Pandemic dataset [21] and then fixed for the duration of the simulation. The home contacts are assumed to repeat their contact with the primary case each day, while each of the work and other contacts encounter the primary case only once during the simulation.

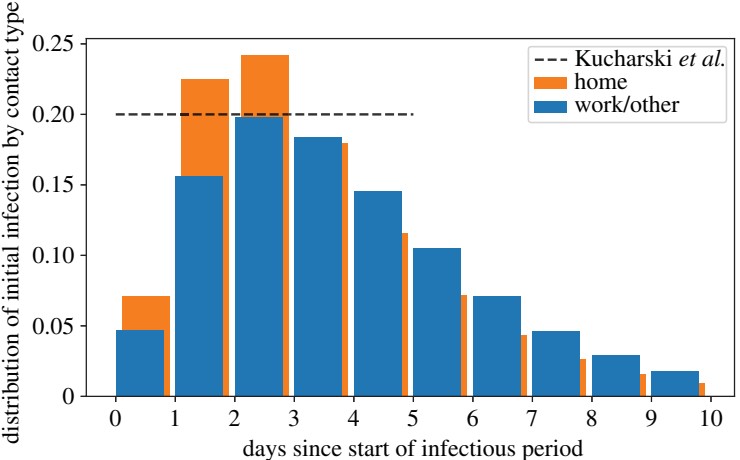

**Figure 6.** Our assumed distribution (blue/orange) of initial exposure to COVID for positive secondary cases. We compare with [6] (black dotted line).

If the primary case is COVID positive, then each of their contacts suffers a risk of infection for each encounter with the primary case (which is drawn independently for each encounter). The risk of infection for a home contact over the simulation period is larger than that of a contact in the work and other categories. Consistent with recent research [20], we model the risk of infection due to an encounter with the primary case as varying over the infectious period, see figure 6 and §A.3.

## A.1. Primary case generation

In this section, we provide a more detailed description of the primary case generation procedure. A `case` is generated as described below. See this in conjunction with the parameter choices given in table 3.

  (i) Sample whether age of `case` is under 18 according to probability `p_under18`.
 (ii) Sample the `presentation` of `case`: symptomatic COVID negative, asymptomatic COVID positive or symptomatic COVID positive.
(iii) If `case` is symptomatic: sample whether they will report symptoms according to the probability `compliance_level`, otherwise, `case` is considered unreported.
 (iv) If `case` reports symptoms: sample the `day_of_reporting` during the 10-day infectious period.
  (v) If `case` reports symptoms: sample whether reporting is done through app with probability `app_coverage`, otherwise, reporting is done manually.

## A.2. Simulating social contact

Once a `case` has been generated following the procedure in §A.1, we simulate their contacts and resulting COVID transmissions, if any. For each simulated contact, we record the *day of first encounter* with primary case during the infectious period, *whether COVID transmission occurred* and, if so, the *day of transmission*. The simulation of contacts and resulting secondary cases is described below. This is in conjunction with parameter choices given in table 4. Note the procedure below applies if `case` is over 18. If `case` is under 18, the procedure is identical, but with `n_work` replaced by `n_school`.

  (i) Sample a participant from the BBC Pandemic dataset, yielding the numbers `n_home`, `n_work` and `n_other` of daily home, work and other contacts the participant had, respectively. We assume `case` has repeated contact with all `n_home` contacts on all 10 days. For work and other contacts, we assume `case` has contact with `n_work` and `n_other` new contacts on all 10 days.
 (ii) If `case` is symptomatic COVID positive, for each contact, sample whether the contact resulted in transmission:
    (a) For home: with probability `sar_home`, the contact becomes infected. If the contact is infected, the day of infection is sampled from the `infection_profile` for home contacts. Our default value for `sar_home` is 0.3.
    (b) For work/other: with probability $p$, contact results in transmission, defining

$$p = 10 \times s \times k,$$

**Table 3.** Modelling choices for case generation.

| parameter | setting | notes |
|---|---|---|
| p_under18 | 0.21 | Probability that the case is under 18 years of age. This affects contact sampling from BBC Pandemic dataset [21]. |
| presentation | $\sim$ Categorical$(\frac{100}{120},\ 0.4 \times \frac{20}{120},\ 0.6 \times \frac{20}{120})$ | Distribution over whether primary case is symptomatic COVID negative, asymptomatic COVID positive or symptomatic COVID positive, respectively. |
| compliance_level | 80% | Proportion of individuals who will comply with government guidelines, report symptoms when they occur, and comply with a quarantine on being traced as a contact. |
| app_coverage | 0.35 | Probability that a primary case has the app and uses it. |
| day_of_reporting | $\sim$ Categorical(0, 0.25, 0.25, 0.2, 0.1, 0.05, 0.05, 0.05, 0.05, 0.00) | Distribution over days that primary case reports COVID-like symptoms and isolates, given that case is symptomatic and decides to report symptoms. Contacts are assumed to be prevented starting from day of reporting and isolation. |

**Table 4.** Modelling choices for transmission.

| parameter | setting | notes |
|---|---|---|
| asymptomatic_factor | 0.5 | Factor by which to reduce probability of transmission when the primary case is asymptomatic following [6]. |
| sar | home: 0.3, work: 0.045, other: 0.045 | Secondary attack rate. Marginal probability a contact (in respective category) is infected by primary case over the duration of the infectious period. |
| infection_profile | Derived from [20]. See figure 6 and §A.3. | Proportional to the probability that contact is infected given day of encounter. Draw independently for each day of contact. |

where $s$ is sar_work or sar_other (we set the default values for both to be the same at 0.045 but one could in theory model differences for different contact types), which is the probability that an average non-household contact becomes infected by the primary case. $k$ is the value of the infection_profile (for work/other contacts) for the day of encounter between case and the contact. $p$ is chosen here as above to maintain both the correct average number of secondary non-household cases and the correct infection_profile.

(iii) If case is asymptomatic COVID positive, perform step (ii), with all secondary attack rates scaled by asymptomatic_factor.

(iv) If case is symptomatic COVID negative, all contacts are uninfected.

Our secondary attack rates (SARs) are chosen to give a base $\mathcal{R}$ of 3.87 when no NPIs are adopted and there is no isolation of a primary case at all for the duration of their infectious period. This is higher than in [6] but consistent with other studies into the initial reproduction number, such as [1]. Moreover, under the assumptions of no repeat contacts, fixed latent period of 3 days and our infection profile (which is described in §A.3), it is possible to calculate for a given $\mathcal{R}$ what the doubling time of pandemic will be (e.g. [25]). Under these assumptions, a base $\mathcal{R} = 3.87$ gives an estimated doubling time of 3.35 days, which is in line with longer estimates of the pre-lockdown doubling time in Europe from a recent study by Ke *et al.* [26]. In practice, factoring into account repeat household contacts will further reduce our estimated doubling time. The assumption of a mean 3-day latent period is consistent with a 5-day incubation period, as suggested by [27], and a 2-day presymptomatic infectious period, as suggested by He *et al.* [20].

Finally, we choose our household SAR and non-household SAR in order to keep the total proportion of infections that are within household in agreement with [6], while maintaining a base $\mathcal{R}$ of 3.87.

## A.3. Modelling secondary infection risk over time

To capture the temporal aspects of TTI policies, it is crucial to model the secondary infection risk over the infectious period, which is how the distribution of the initial infection of secondary contacts varies over the length of time that a primary case is infectious. In order to do this, we need to understand the timeline of infection for the primary case. When a case is infected, it is widely supposed for COVID that there is an initial latent period of around 3 days, when the case is neither infectious nor symptomatic [20]. For our purposes, the latent period of the primary case is not important, as a result, we only start modelling primary cases once they become infectious, like in [6]. However, it is important to note that we will be interested in the latent period of the positive secondary contacts, in order to count the fraction of the secondary contact's infectious period before they are traced by our TTI policies, as described in appendix D.

At this point, in order to capture the secondary infection risk over time we need to model the infectiousness profile, which is the distribution of relative infectiousness of the primary case over the course of his/her infectious period. The infectiousness profile often assumes $t = 0$ to be the time when the infector develops symptoms. However, we will set $t = 0$ to be the start of the infectious period, because that is the point from which our model starts simulating cases and contacts.

One way to approximate the infectiousness profile would be to collect viral shedding data of the primary case over the course of the infectious period. For our purposes, the infectiousness profile is useful as it is exactly the distribution of when positive secondary contacts are infected for contacts who are only met once during the infectious period, and if there is a constant number of such new contacts each day. In our model, this is the case for work and other contacts, and hence the infectiousness profile is exactly the secondary infection risk distribution for our non-household contacts.

The recent findings in [20] inferred that the infectiousness profile should be skewed towards early transmission, with 44% (95% confidence interval, 25%–69%) presymptomatic transmission. Indeed, He *et al.* [20] fit a Gamma distribution to the inferred infectiousness profile using data of 77 known transmission pairs, with shape parameter 2.11 and rate parameter 0.69.

One has to be careful of the distinction between relative infectiousness of primary cases, measured in terms of viral shedding and the relative likelihood of when a repeated secondary contact, such as a household contact who goes on to be infected, was initially infected with COVID. For example, Uniform infectiousness over the infectious period will correspond to a geometric distribution for the initial infection of a secondary contact who the primary case meets everyday. Because most of the pairs of data that [20] fit were repeated contacts, this means that the Gamma distribution they fit has undue bias towards early and/or presymptomatic infection. In order to account for this bias, we will use shape parameter 2.8 and rate parameter 0.69 to model our infectiousness profile. That is to say, the mean time of infection for secondary non-household contacts will be delayed by 1 day relative to the fitted Gamma distribution in [20]. Our model uses a discretized version of this fitted Gamma distribution for the infectiousness profile of our primary cases over the infectious period. We present a sensitivity analysis for our assumptions in appendix F.

For household contacts, because our model assumes that home contacts are met everyday as opposed to work/other daily contacts who are different contacts each day, this implies that the distribution of when a home secondary contact was infected is skewed more towards early transmission, as can be seen in figure 6. To sample from this distribution, we first sample if a household contact was infected at all by the primary case over the infectious period, with Bernoulli probability `sar_home`. Then, for infected household contacts, the day on which they were infected can be sampled by flipping a coin each day of the infectious period independently with probability heads equal to the infectiousness profile on that day. The process stops either when we get first heads or when we get to the end of the infectious period. If we get to the end of the infectious period without a heads, we return to the first day of the infectious period and repeat this process of coin flipping until we reach a first heads, which will almost surely occur eventually. The day on which the first heads lands is the day of transmission.

**Table 5.** Parameters for different NPI severity levels.

| attribute | S5 | S4 | S3 | S2 | S1 | notes |
|---|---|---|---|---|---|---|
| work_from_home_ proportion | 65% | 55% | 45% | 25% | 0% | The proportion of the population not going into their regular workplace. 35% of people are not going into work as usual, and 45% are working from home (as of 7 May 2020, [29]). |
| school_from_home_ proportion | 100% | 100% | 50% | 0% | 0% | The proportion of school-aged children not going into schools. |
| max_other_contacts | 1 | 4 | 10 | 20 | - | A hard limit placed on the number of non-home, non-work contacts a person has per day. |
| work_met_before_ proportion | 79% | 79% | 79% | 79% | 79% | The average proportion of work contacts a person has met before, allowing them to manually trace a contact. Taken from [21]. |
| school_met_before_ proportion | 90% | 90% | 90% | 90% | 90% | The average proportion of school contacts a person has met before, allowing them to manually trace a contact. Taken from [21]. |
| other_met_before_ proportion | 100% | 100% | 90% | 75% | 52% | The average proportion of other contacts a person has met before, allowing them to manually trace a contact. Taken proportion from [21], adjusted for lockdown. |

We choose to discretize over an infectious period of 10 days compared with only a 5-day period in [6]. This is in line with the assumption that the presymptomatic period lasts for approximately 2–3 days on average and that infectivity is much lower after the first week of symptom onset, as shown in [28]. We assume that the infectiousness profile is identical across primary cases, including asymptomatic cases, following the discretized Gamma distribution.

# Appendix B. Scenarios of other non-pharmaceutical interventions

We consider the impact of TTI strategies in the context of a range of scenarios with varying stringencies for other non-pharmaceutical interventions (NPIs). These are:

  (i) S5 - Lockdown (up to 9 May 2020)
 (ii) S4 - Slightly relaxed work and social restrictions
(iii) S3 - Moderately relaxed work and social restrictions
 (iv) S2 - Strongly relaxed work and social restrictions
  (v) S1 - No social restrictions, but quarantining of symptomatic households remains in place.

Each of these scenarios modify a further set of parameters that influence the effectiveness of a TTI strategy. These are listed in table 5. These parameters affect two factors: the reduction in number of contacts a given person has due to social distancing measure, and how likely it is to trace a particular contact during the stages of lockdown.

The `work_from_home_proportion` and `school_from_home_proportion` govern the fraction of adults and children not going to their normal place of work/school. Each primary case has these probabilities of not going to work/school, in which case we remove all of the work/school contacts for that individual.

The `max_other_contacts` parameter governs the effects of general social distancing, and sets the maximum other contacts of a primary case each day.

# Appendix C. Test-trace-isolate strategies

In this section, we describe how we model different TTI strategies.

Three types of case pass through the TTI funnel:

— Symptomatic COVID cases.
— Asymptomatic COVID cases.
— Cases presenting with COVID-like symptoms, but COVID negative.

From the perspective of TTI, asymptomatic COVID cases are invisible and contribute to an unavoidable baseline $\mathcal{R}$ that only other NPIs, like physical distancing measures, can help reduce. On the other hand, cases that present symptoms similar to COVID, but are not COVID positive, can only contribute to an increase in the cost of TTI and have no impact on $\mathcal{R}$.

We consider three options for contact tracing, with decisions made around when to isolate the contacts of an individual, and whether to test contacts of positive cases, or just quarantine them. These can be summarized as:

(i) Isolate all contacts on a primary case presenting with symptoms.
(ii) Isolate household contacts immediately, and isolate non-household contacts upon positive COVID test.
(iii) Isolate household contacts immediately, and isolate non-household contacts upon positive COVID test. Test these contacts for COVID.

We consider test-based TTI as the baseline strategy. Test-based TTI with contact testing trades off using extra tests to get contacts who are not infected out of quarantine quicker. Symptom-based TTI isolates contacts faster, to allow them less chances to cause new infections, but will cause the quarantining of many additional people from cases presenting like COVID that are not in fact COVID. Flow charts for these strategies are shown in figures 7–9.

We suppose that symptomatic cases have two routes via which they can enter the TTI system: by submitting symptoms manually to the NHS or by using the NHSx app. The proportion of the population with the app is governed by the `app_coverage` parameter, and is sampled independently for each case and contact. If a case is sampled as having the app, they are assumed to use it to report symptoms and allow the use of app tracking. We assume that some cases do not submit symptoms, due to either not taking them serious enough, or not complying with the guidelines. In our model, this is governed by the `policy_compliance` parameter. This compliance also applies to contacts traced, sampled independently for each case and contact.

If the symptomatic primary case enters the TTI system they are assumed to follow the government guidelines and isolate at home alongside the rest of their household. We make this assumption, as to have been tested, a case would have to have already voluntarily reported symptoms. In the case where no contact tracing is being performed, we assume that no tests are performed on symptomatic individual who reports symptoms (or would have reported symptoms), but they do still isolate at home, as is the current situation.

There is a delay in getting test results back, and this delay is governed by the `test_delay` parameter in our model. Given that there is a significant delay between testing an individual and getting a result, a choice is presented here: do we perform contact tracing before or after the test results? Both of these options are explored.

Upon initiating contact tracing, a case's contacts are traced manually, and also via the app if the case has the app. App-based tracing succeeds with a probability governed by the `app_coverage` parameter, sampled independently for each contact. Manual tracing succeeds if a person is able to identify a contact and provide details of them. Using data from [21], we calculate the likelihood a person has met a contact before at work, at school and elsewhere (denoted as `other` category). These are encoded in the `work_met_before_proportion`, `school_met_before_proportion` and `other_met_before_proportion`. These probabilities are sampled independently for each contact to see if manual contact tracing is effective. We also assume a fixed time delay in contacting a person's contact from the point at which we decided to quarantine them. These delays are encoded in

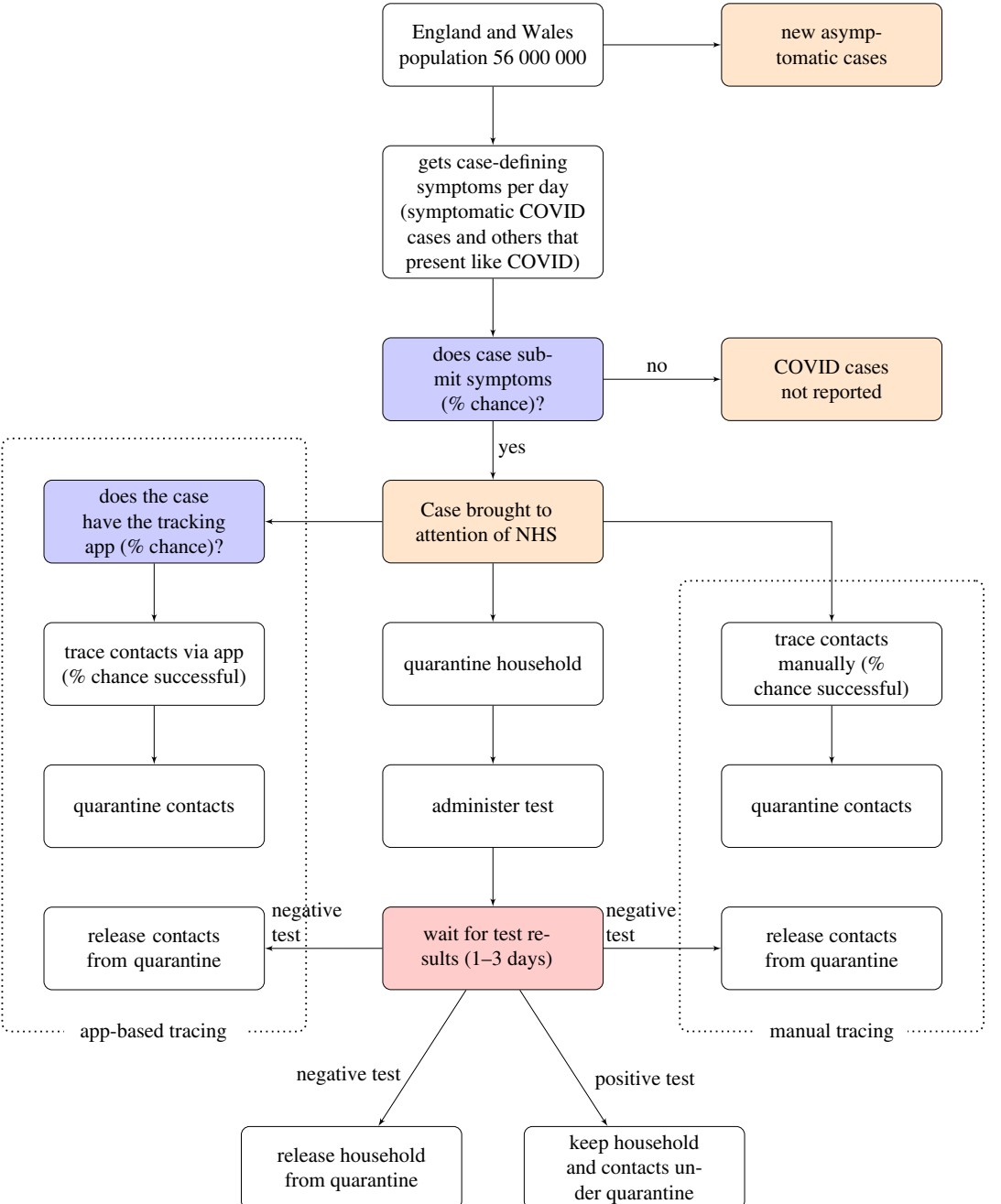

**Figure 7.** Diagram of an individual passing through a symptom-based TTI system.

the `manual_trace_delay` and `app_trace_delay`. Once a person is traced, they comply with the quarantine with probability defined by the `policy_compliance` parameter.

For non-household contacts, once they are traced by the TTI system (either before or after the test results), they are advised to isolate themselves at home for 14 days. If they have isolated on notification of symptoms of a primary case, and if that person is tested and comes back negative, the contacts are released from quarantine. If they have quarantined after notification of a positive test of the primary case, they remain in the quarantine the full 14 days. We consider also an additional scenario where the contacts of an individual are tested on a positive test of a primary case. In order to avoid further simulation, we suppose that a fraction of positive secondary contacts will go on to become COVID positive in line with our assumptions in §A.2. In this case, if a contact tests negative they may be released from quarantine early.

A full table of the parameters used in the TTI strategies can be found in table 6. We choose realistic defaults for these in line with current situation in the UK, and perform sensitivity analysis over them later to see either the effect of expending effort to improve these parameters, or to account for errors in our assumptions.

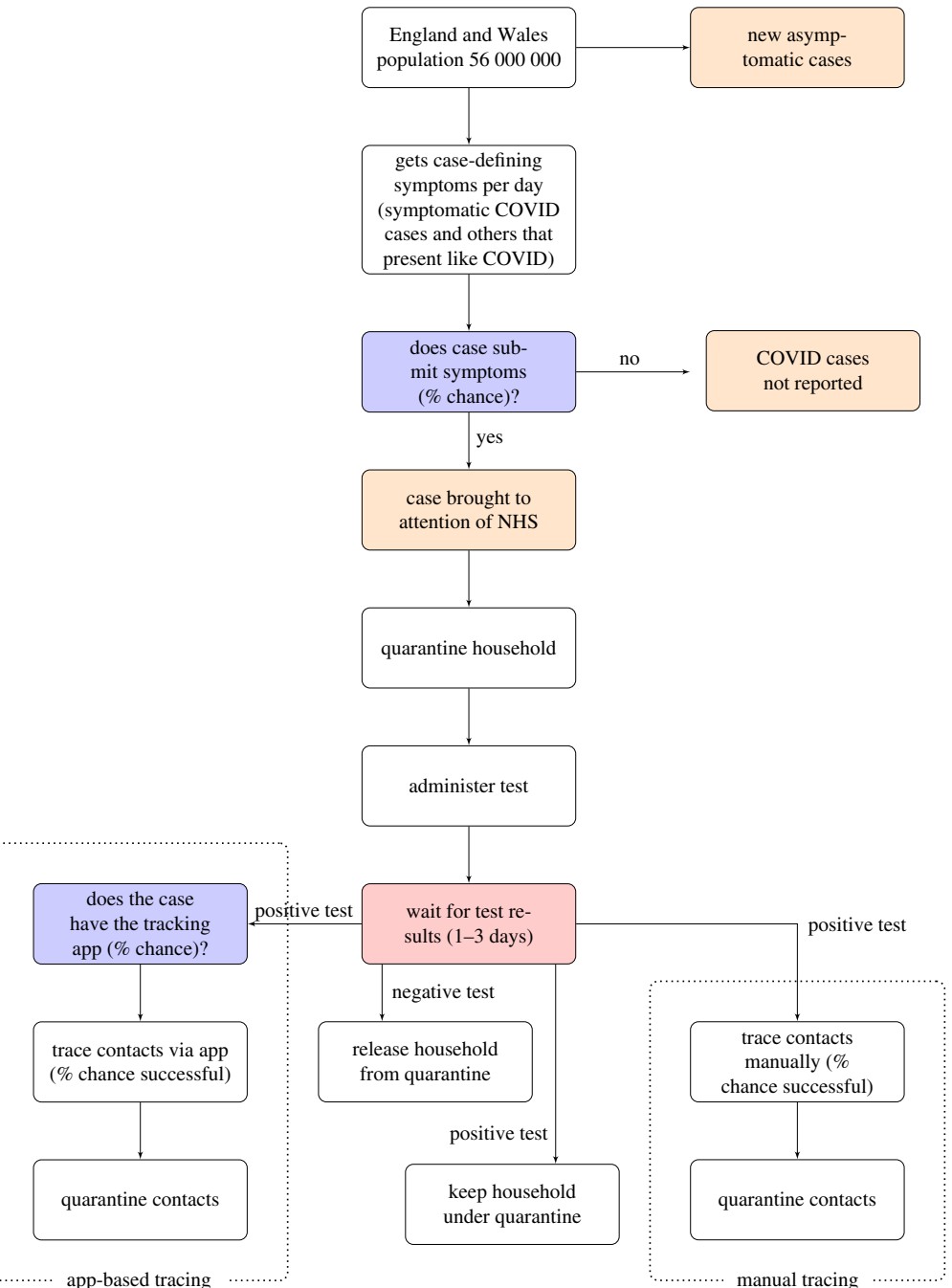

**Figure 8.** Diagram of an individual passing through a test-based TTI system.

# Appendix D. Metrics to evaluate test-trace-isolate strategies

One key goal of this study is to compare the effectiveness and cost of various TTI strategies.

In terms of cost, we report results for the number of: manual traces, tests required, and person-days spent in quarantine.

In terms of effectiveness, the main metric we use is the effective $\mathcal{R}$ number that our strategies result in. It is possible during contact tracing that we do not manage to notify a contact that they are an infection before they become infectious. We count these infectious contacts partially as follows: for each such secondary contact, we add a fraction between 0 and 1 to the primary case's individual $\mathcal{R}$ number that is:

— 0 if the infection was prevented via isolation of the primary case or other NPIs,
— 1 if the TTI system did not trace the infected contact, including if the primary case was asymptomatic or did not report symptoms, and

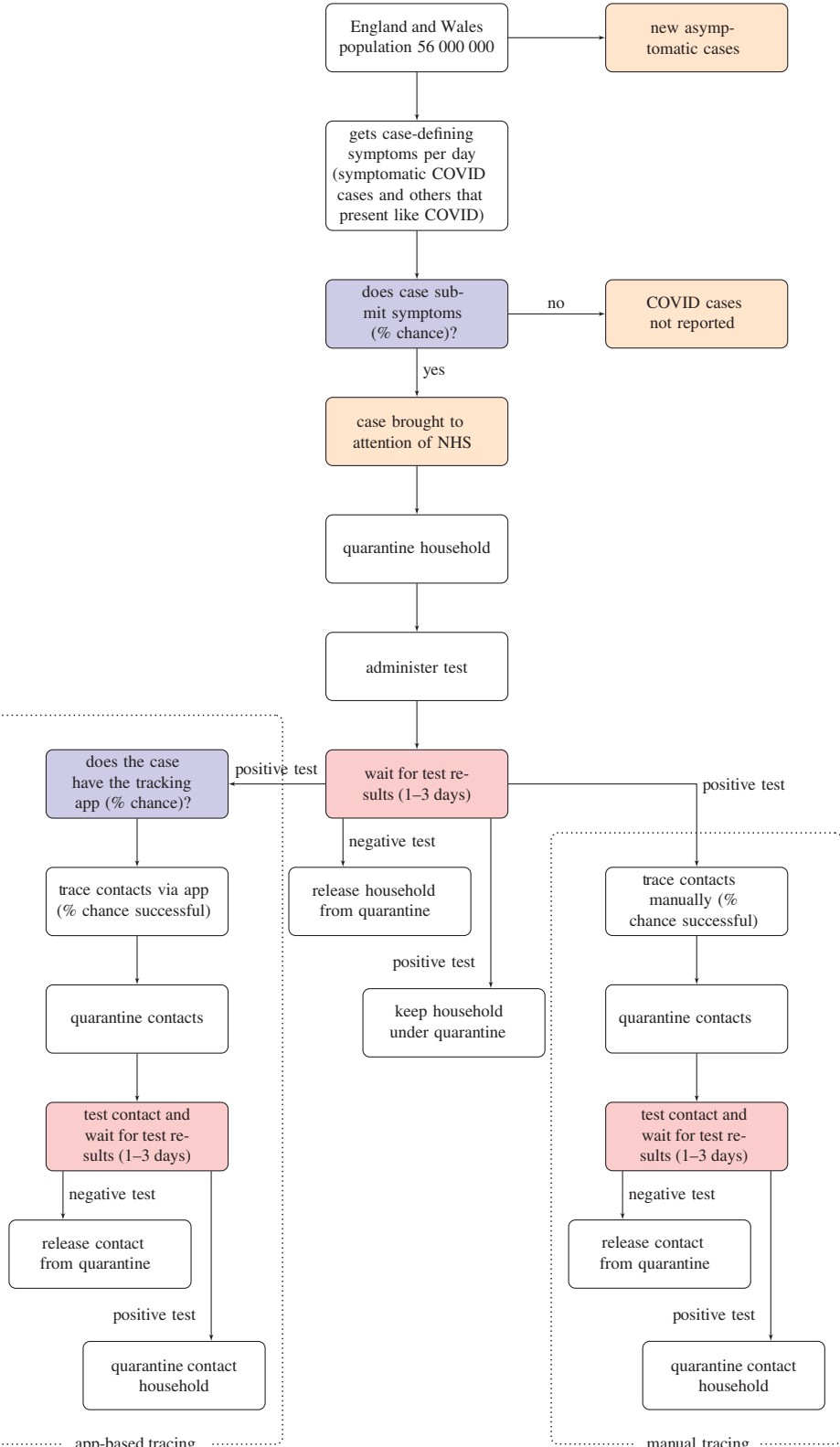

**Figure 9.** Diagram of an individual passing through a test-based TTI system with testing of contacts.

— the fraction of the secondary contact's infectious period before they were traced and quarantined by the TTI system, weighted by the infectiousness profile as described in appendix A.3. We assume a default 3 days for the pre-infectious latent period of the secondary contact as consistent with [20]. This fraction reflects the reduction in transmission as a result of contact tracing and quarantining.

**Table 6.** Parameters describing pinch points, policy features and population characteristics used when simulating TTI strategies.

| attribute | current setting | notes |
|---|---|---|
| app_coverage | 35% | The proportion of the population who take up using the NHSx tracing app |
| test_delay | 2 days | Delay between test and result, assumed 0 days in [6] |
| manual_trace_delay | 1 day | Delay between a test result and notifying contacts manually, assumed 0 days in [6] |
| app_trace_delay | 0 days | Delay between a test result and notifying contacts via app, assumed 0 days in [6] |
| policy_compliance | 80% | Proportion of individuals who will comply with the government guidelines, report symptoms when they occur, and comply with a quarantine on being traced as a contact. |
| quarantine_length | 14 days | Length of quarantine |

As for the primary case, there is some chance for the contact to ignore the quarantine request, in which case this is counted as 1.

## D.1. Confidence intervals

In our simulations, each primary case is sampled independently at random from the same distribution (with the same probability of being COVID positive, symptomatic and so on). The results are averaged over the runs of the simulation and scaled to the correct population size. For instance, to compute the effective $\mathcal{R}$ due to 100k COVID positive primary cases, we average the infections due to each of these primary cases and multiply by 100k. This is a Monte Carlo approximation and the confidence intervals are computed accordingly: the 95% confidence interval of metric $m$ that is averaged over $N$ trials with (empirical) standard deviation $\hat{\sigma}$ is $m \pm 1.96(\hat{\sigma}/\sqrt{N})$. Returning to the case of effective $\mathcal{R}$, if the mean and standard deviation of the number of secondary infections due to the 100k primary cases are $\hat{\mathcal{R}}$ and $\hat{\sigma}$, respectively, then the 95% confidence interval is given by $\hat{\mathcal{R}} \pm 1.96 \times \hat{\sigma}/\sqrt{100\text{k}}$.

# Appendix E. Key parameters for the effectiveness of test-trace-isolate strategies

This appendix contains additional results investigating the effect of various parameters on the efficacy of TTI strategies, expanding the results in the main text.

In figure 10, we consider variation in application uptake and compliance within the population for all TTI strategies. In the left-hand column, we examine the effect of increasing app uptake on the effective $\mathcal{R}$ of TTI strategies, we see that the app is most significant as the NPI severity level is decreased. In the right-hand column, we observe that compliance with requests to isolate and to contact trace is a highly significant factor in the effectiveness of TTI strategies.

Figures 11–15, bottom left show the proportion of ongoing transmission that is prevented by TTI under the test-based TTI strategy, and how this varies with delays in testing and manual tracing on the proportion of ongoing transmissions prevented by test-based TTI. Note the increase in effectiveness of TTI as the total delay decreases.

Figures 11–15 top left likewise show how this varies with uptake of the app. We observe that the proportion of ongoing transmissions prevented by TTI increases as application uptake increases, especially for lower severity levels. This is due to more effective tracing and isolating of the contacts of the primary case. The variation between severity levels is due to variation in numbers of contacts for primary cases (due to social distancing).

Finally, figures 11–15 top right shows how the proportion of ongoing transmission that is prevented by test-based TTI varies with public compliance in the context of various levels NPI severity. We see that public compliance significantly improves TTI effectiveness. Under lower severity NPIs, there is more social contact, so compliance becomes more significant.

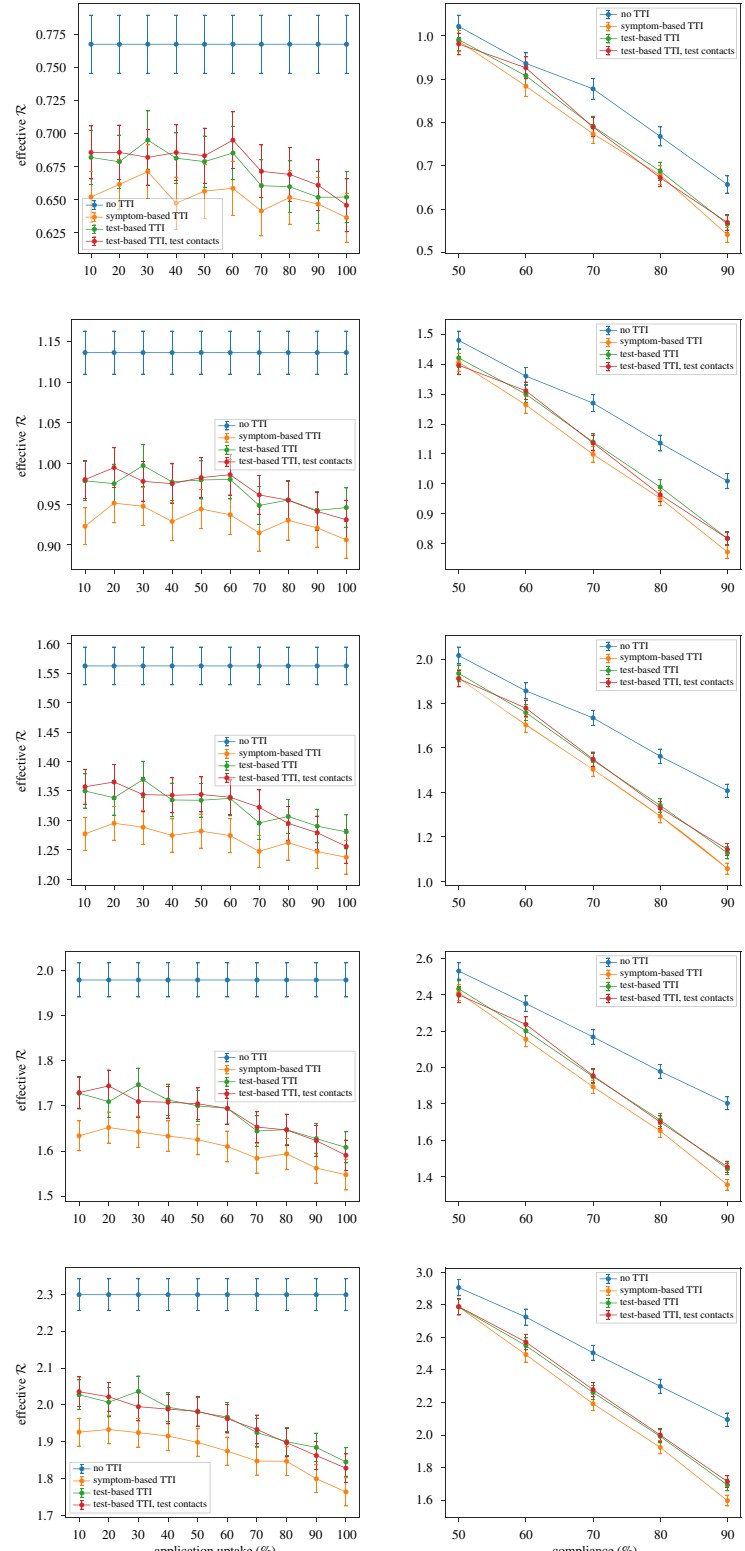

**Figure 10.** *Left*: Effect of varying the app uptake on effective $\mathcal{R}$. *Right*: Effect of varying the policy compliance on effective $\mathcal{R}$. *Top to Bottom*: S5 down to S1.

# Appendix F. Sensitivity analysis

This appendix investigates the sensitivity of our model to a number of assumptions we have made about the number of COVID cases in the UK, and the typical timeline of a COVID infection.

Figure 16 shows the variation in the number of tests needed for TTI strategies under changes in the number of COVID positive cases and the number of COVID negative cases with COVID-like symptoms. Our results are

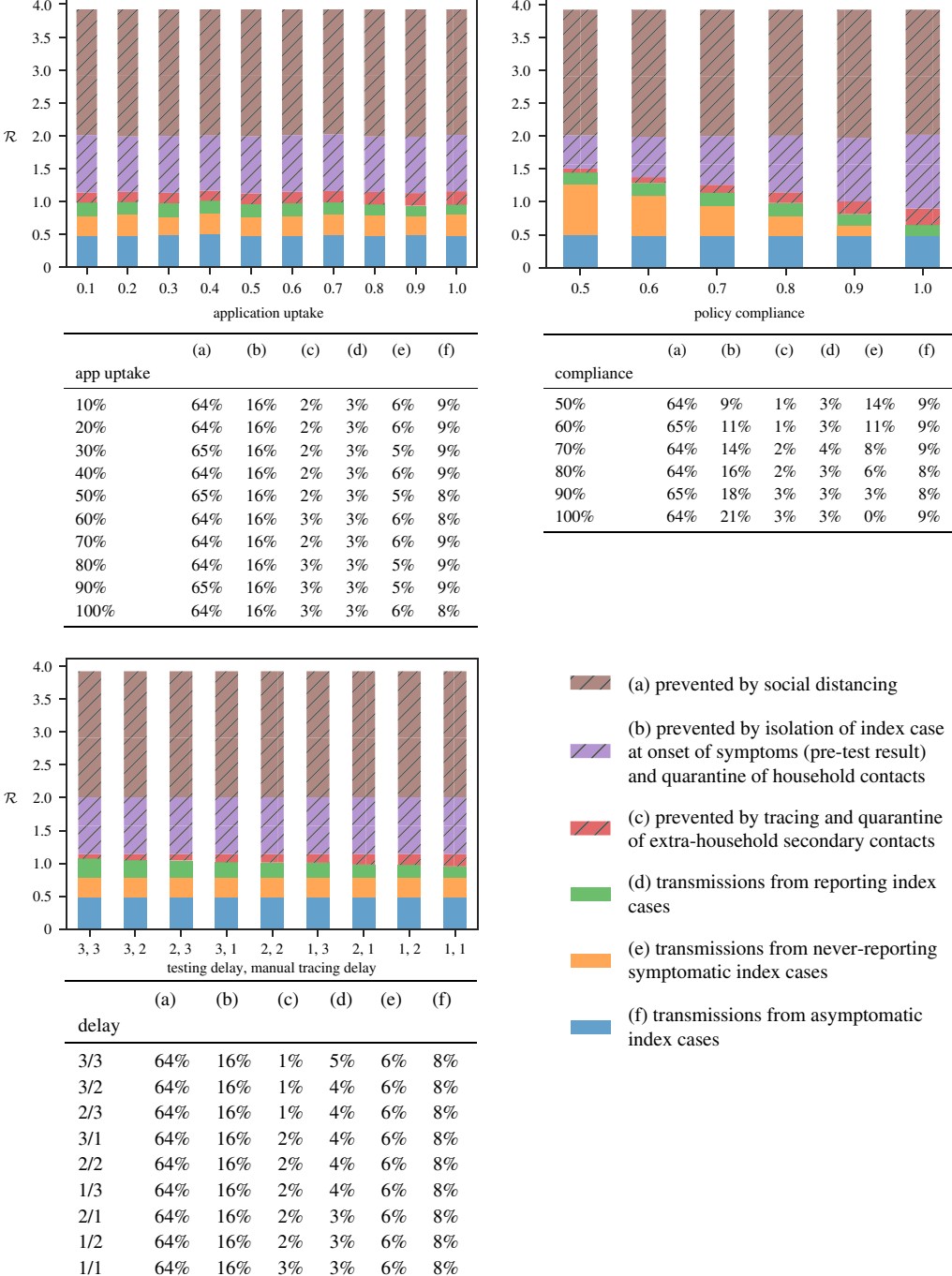

**Figure 11.** The impact on the percentage of ongoing transmission reduced by TTI of changing the application uptake rate, the policy compliance rate, and reducing delays for testing and manual tracing for the S5 severity levels and using the test-based TTI strategy.

consistent across these case for all severity levels S5–S1. Across the cases, the increase in tests needed for the TTI strategies is due to an increased number of symptomatic individuals.

Figure 17 considers variation in the effective $\mathcal{R}$ of TTI strategies under changes in the timeline of a COVID infection. Again we observe that, within measurement error, our results are consistent across the cases. In the left-hand column, the effective $\mathcal{R}$ of TTI strategies is reduced if the primary case is most infectious later in the lifetime of their infection; here, the primary case causes fewer secondary infections before isolation. In the middle column, we see a similar phenomena in reverse: the effective $\mathcal{R}$ increases with the time taken for a primary to report symptoms and isolate. In the right-hand column, we see that the effective $\mathcal{R}$ is relatively insensitive to the length of the latent period. Increased latent period allows TTI more time to track contacts of a primary case before they become infectious (if infected).[2]

---

[2]Recall that the simulation begins at the *end* of the latent period of the primary case.

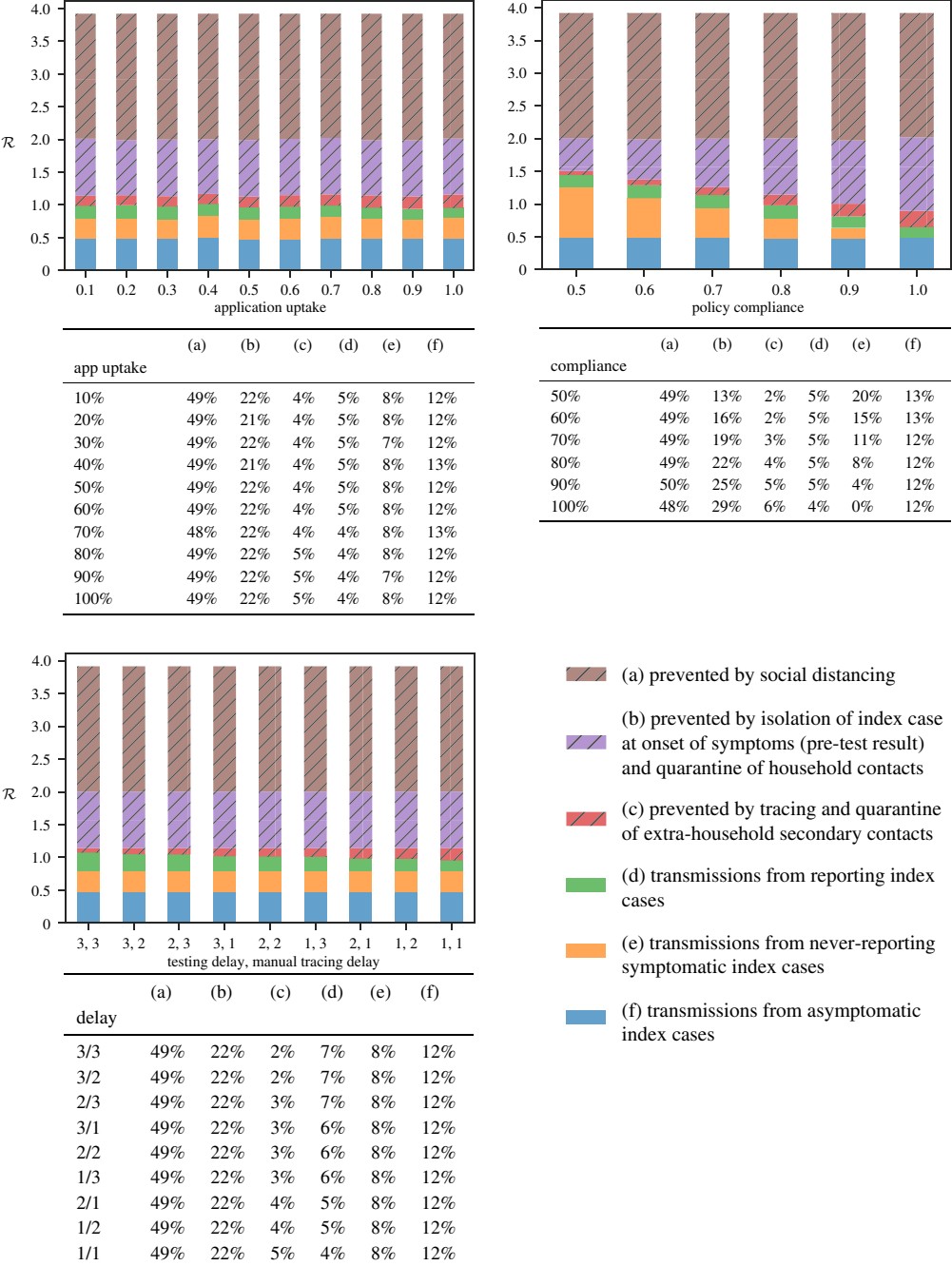

**Figure 12.** The impact on the percentage of ongoing transmission reduced by TTI of changing the application uptake rate, the policy compliance rate and reducing delays for testing and manual tracing for the S4 severity levels and using the test-based TTI strategy.

# Appendix G. Relationship between number of infections and $\mathcal{R}$

Figure 18 displays the exponential growth over 30 days of an epidemic seeded with 20k primary infections, for our assumptions of infectiousness profile and latent period, for different settings of $\mathcal{R} > 1$. We calculated the growth rate using eqn 2.7 of [25]. Note that our calculations assume that the infectiousness profile and latent period are constant no matter the value of $\mathcal{R}$, and also this calculation does not consider repeated contacts such as household contacts. Moreover, figure 18 assumes that the level of $\mathcal{R}$ is fixed over time, which may not reflect reality. However, figure 18 does highlight the fact that even small changes to $\mathcal{R}$ that result from TTI can be crucial in combating the spread of COVID.

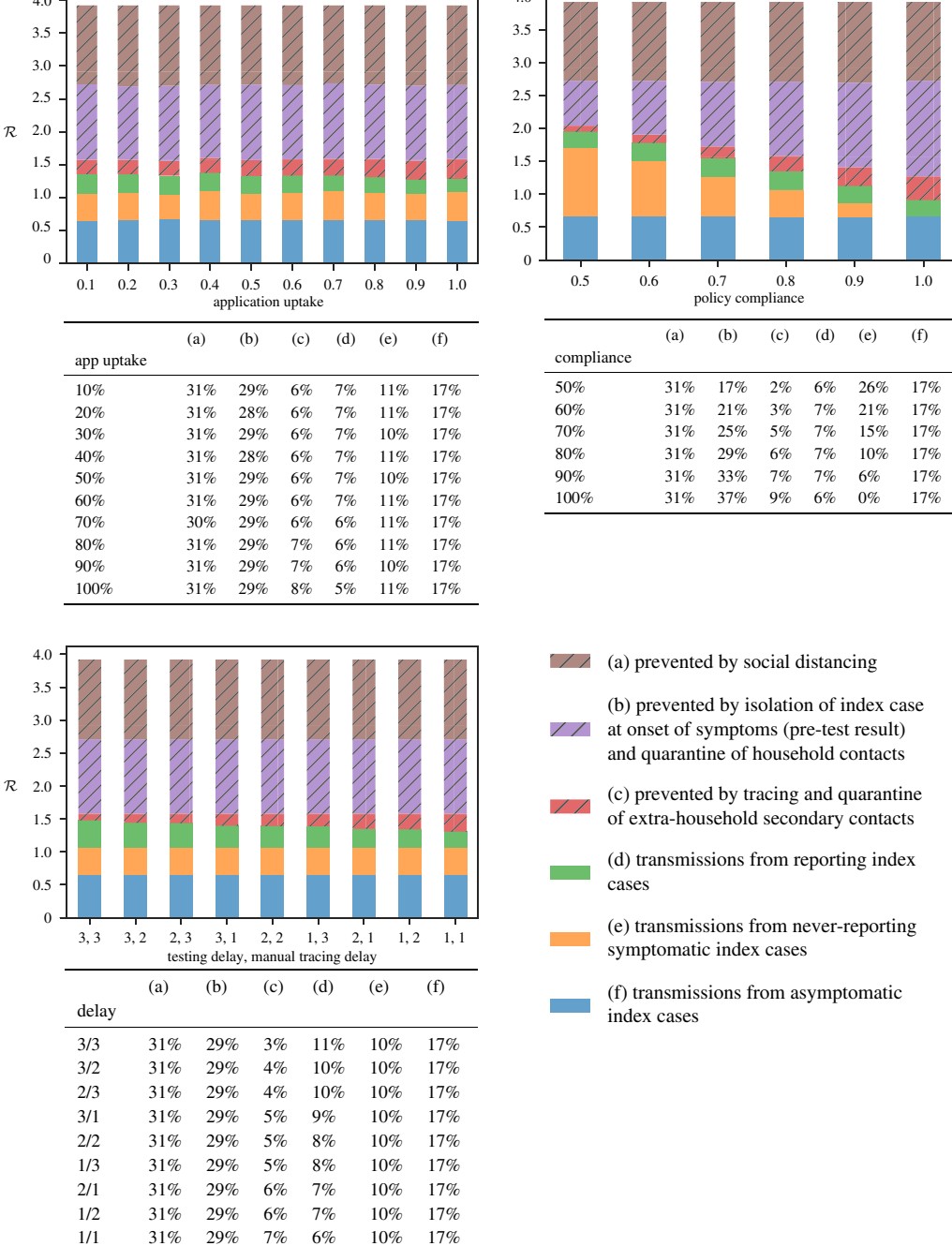

**Figure 13.** The impact on the percentage of ongoing transmission reduced by TTI of changing the application uptake rate, the policy compliance rate and reducing delays for testing and manual tracing for the S3 severity levels and using the test-based TTI strategy.

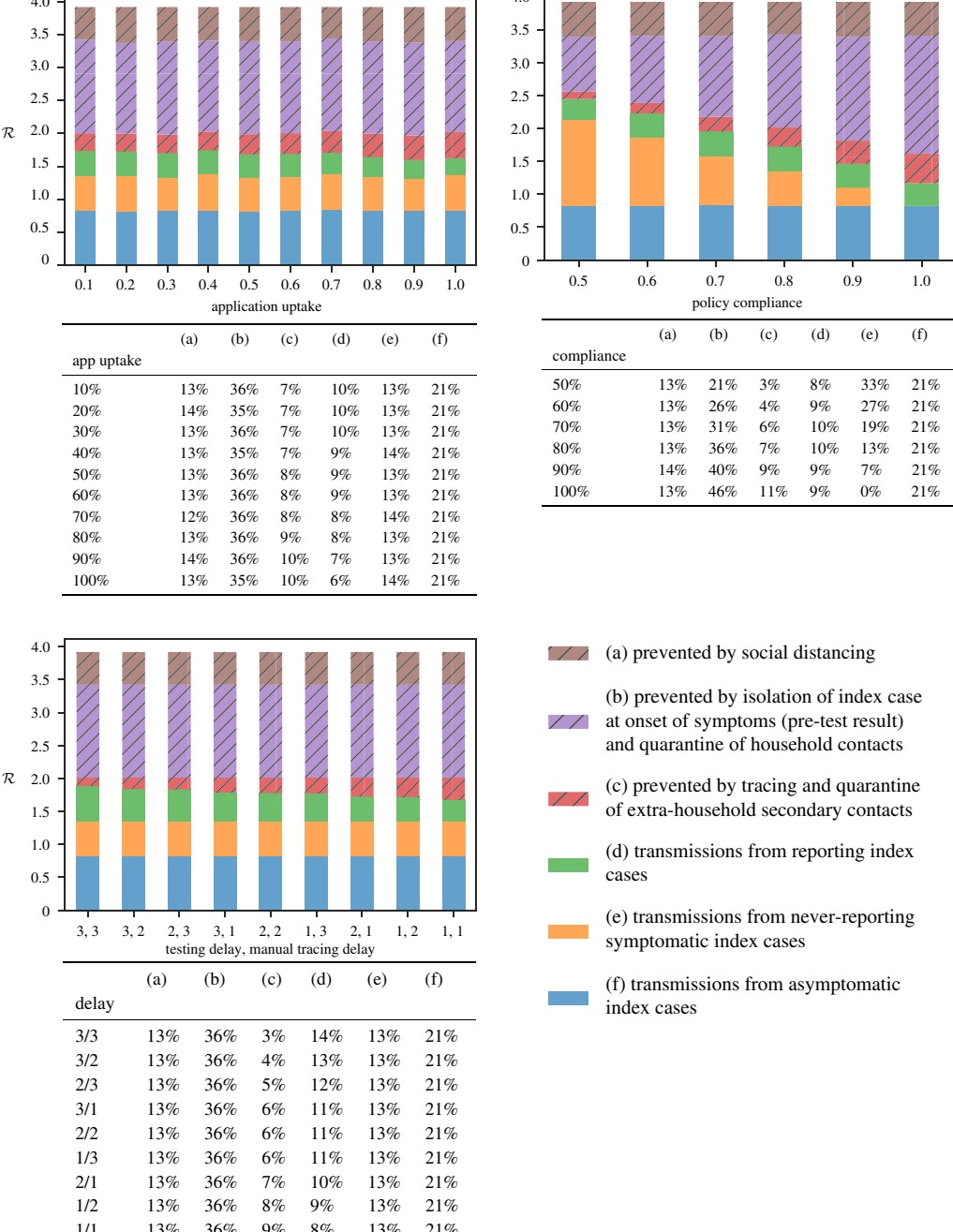

**Figure 14.** The impact on the percentage of ongoing transmission reduced by TTI of changing the application uptake rate, the policy compliance rate and reducing delays for testing and manual tracing for the S2 severity levels and using the test-based TTI strategy.

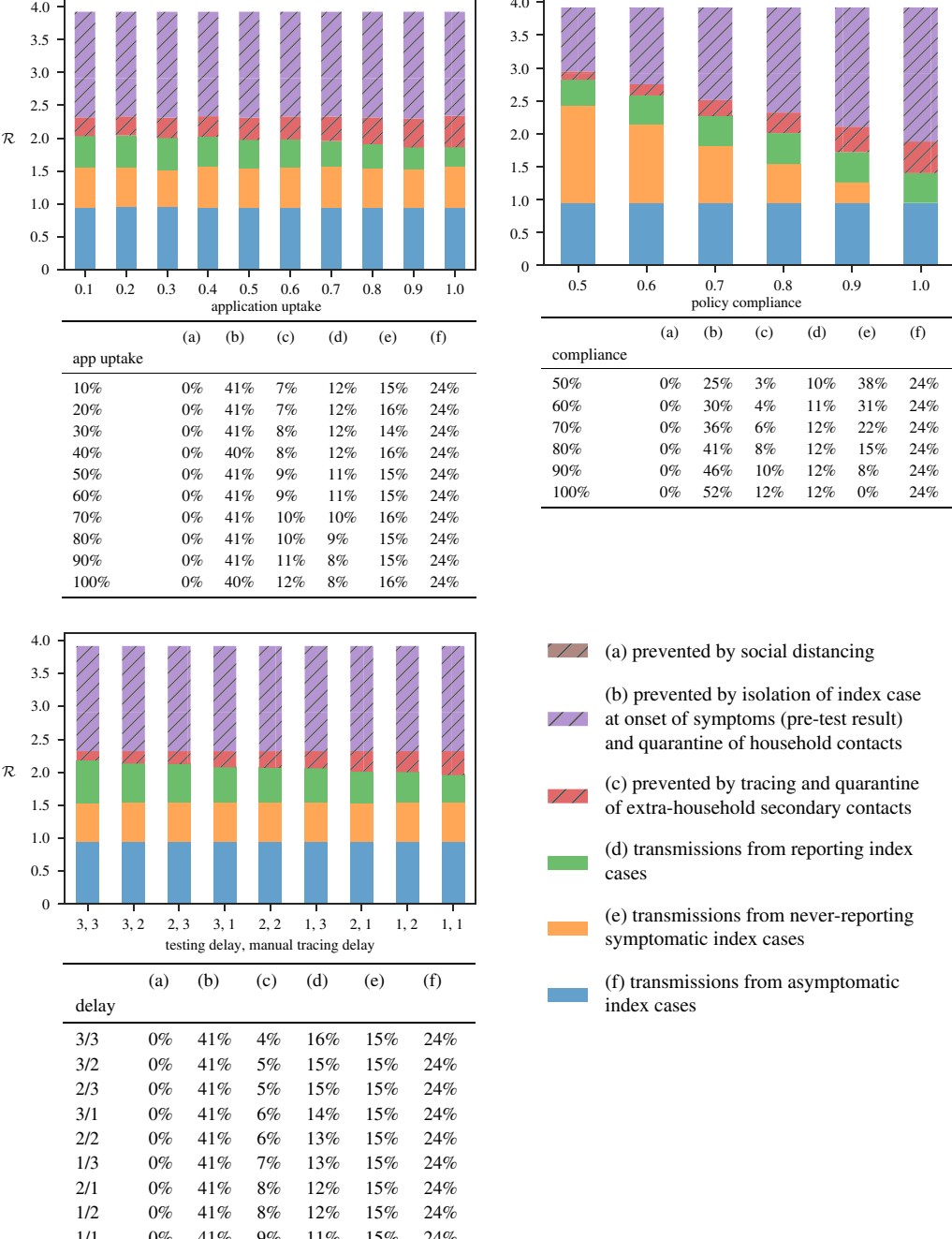

**Figure 15.** The impact on the percentage of ongoing transmission reduced by TTI of changing the application uptake rate, the policy compliance rate and reducing delays for testing and manual tracing for the S1 severity levels and using the test-based TTI strategy.

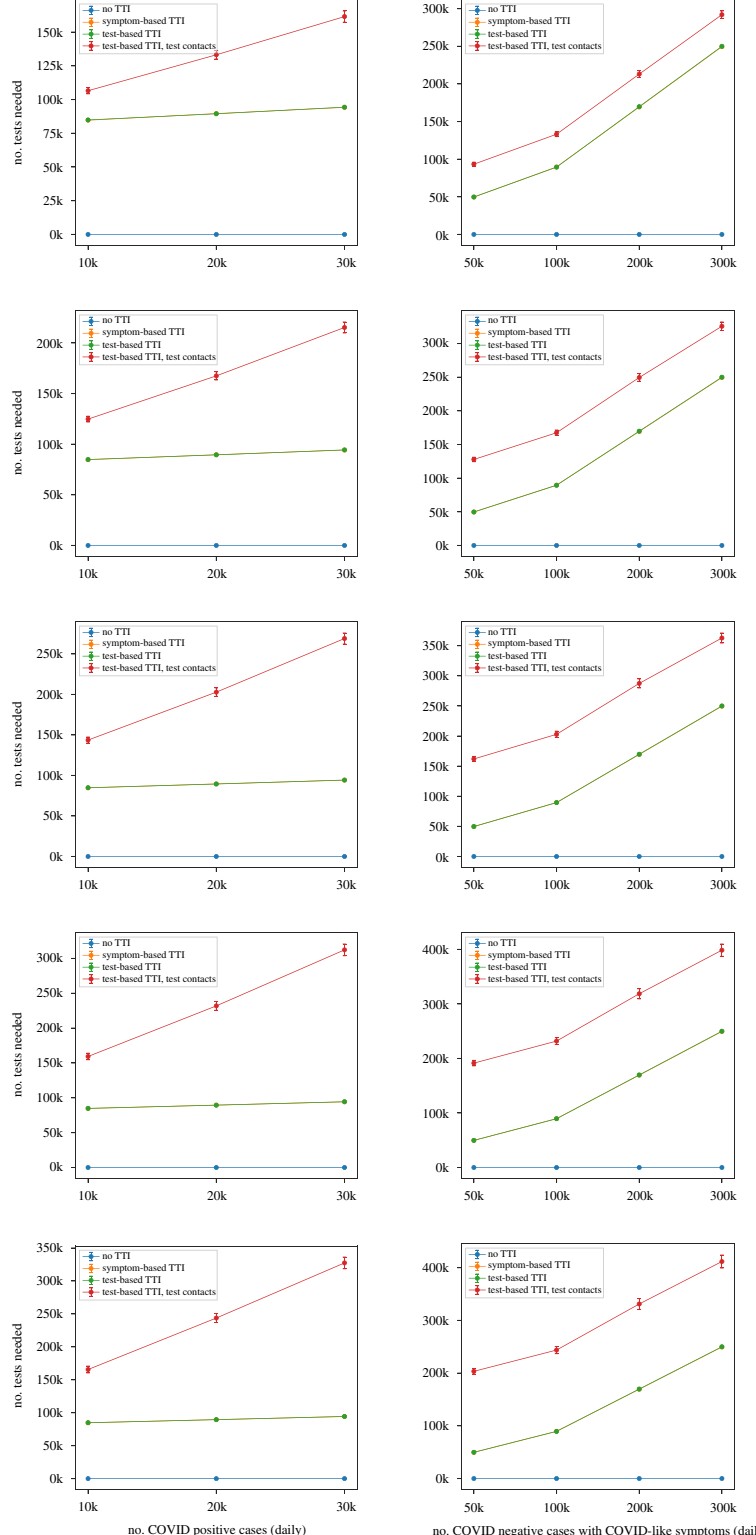

**Figure 16.** *Left*: Effect of varying number of new COVID cases per day on the number of tests needed. *Right*: Effect of varying the number of COVID negative cases with COVID-like symptoms on the number of tests needed. The number of tests required for symptom-based TTI is the same as for test-based TTI, as we assume all primary cases are tested. *Top to Bottom*: S5 down to S1.

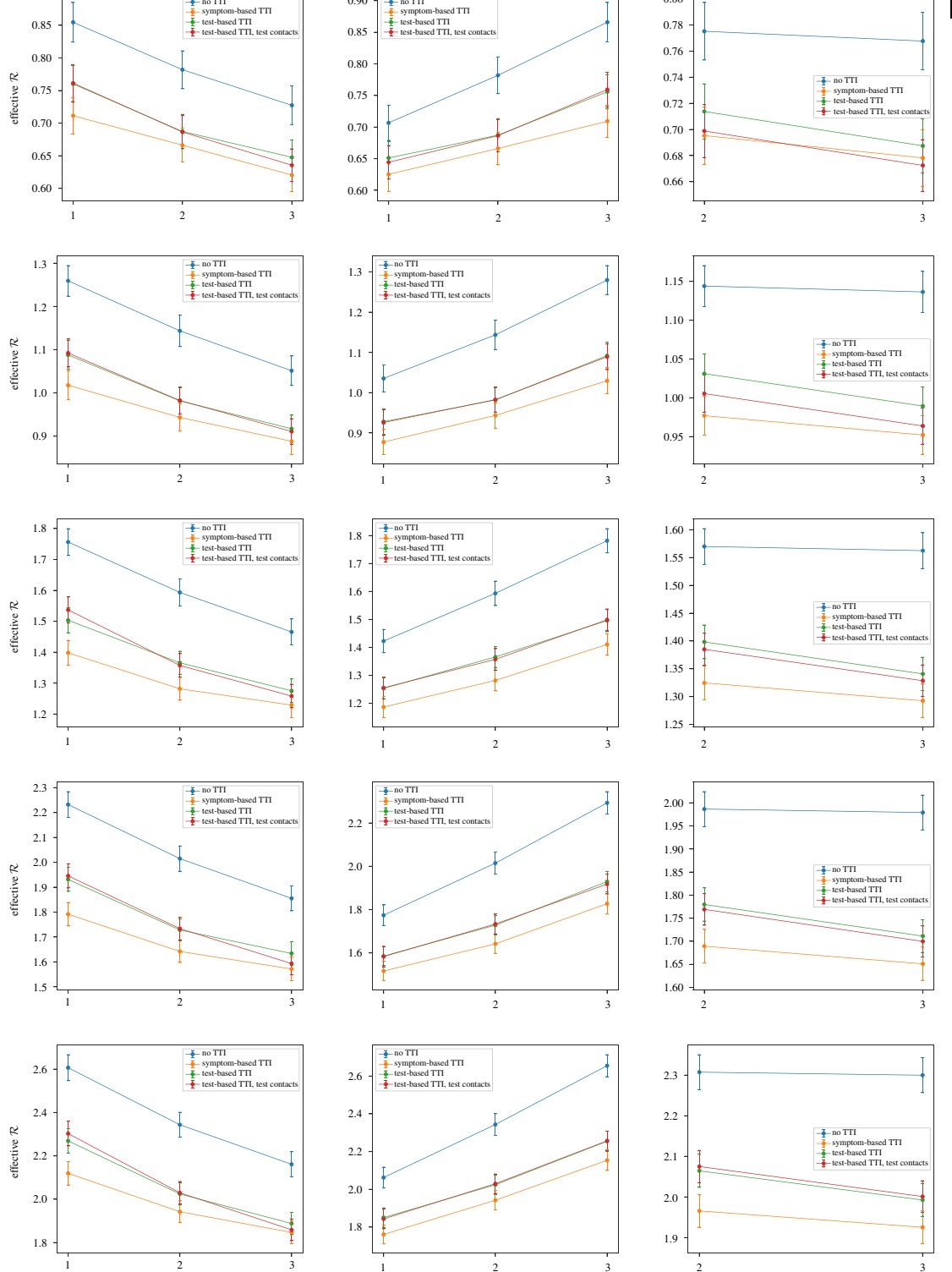

**Figure 17.** *Left*: Effect of varying the most infectious day of the infectious period for COVID cases on $\mathcal{R}$. *Middle*: Effect of varying expected day of symptom reporting (measured from after the end of the latent period) on $\mathcal{R}$. *Right*: Effect of varying the latent period on $\mathcal{R}$. *Top to Bottom*: S5 down to S1.

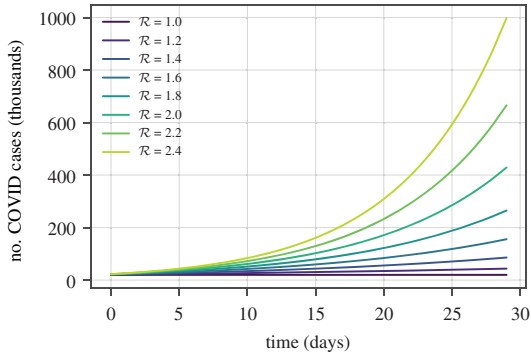

**Figure 18.** Relationship between expected number of new infections and $\mathcal{R}$ over time.

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
