## [Peer Review File · Royal Society Open Science]

Review History

RSOS-201491.R0 (Original submission)

Review form: Reviewer 1 (Seth Flaxman)

Is the manuscript scientifically sound in its present form?

Yes

Are the interpretations and conclusions justified by the results?

Yes

Is the language acceptable?

Yes

Do you have any ethical concerns with this paper?

No

Have you any concerns about statistical analyses in this paper?

No

Recommendation?

Accept with minor revision (please list in comments)

Comments to the Author(s)

Abstract:

This is a timely and compelling piece of work. It is well-presented, including both the strengths, and limitations. I have a number of minor comments below, all of which should be straightforward to address.

- 1) "We further show that adding contact tracing for extra-household contacts of confirmed cases to this broader package" -- clarify what this means
- 2) " we use contact simulation model " -> " we use *a* contact simulation model "
- 3) For your discussion, worth citing <https://www.medrxiv.org/content/10.1101/2020.09.15.20191957v1.full.pdf> in the context of your assumption of 80% compliance. Since it looks pretty easy to rerun your simulations, you could take some of the parameters from that study and do an extra analysis showing that TTI is (presumably) ineffective based on their study.
- 4) Section 2, paragraph 1, missing reference
- 5) [10] should be updated to the Nature publication. The 20k number seems perfectly reasonable, but you should clarify exactly where it came from---neither report 13 (currently cited) or Flaxman et al (Nature) extend to 10 May. For context, the most recent ONS report says upper bound 8,300 cases per day in England. Scaling that up to the UK gives about 10k cases per day. (Returning to this comment, you'll still need to cite [10] for the 3.87 R0 you use throughout.)
- 6) The phrasing of "We consider five scenarios of combinations of NPIs with varying stringency levels." confused me, especially as it appears just below Table 1. Perhaps present TTI strategies first? Either way, rephrase that sentence to make it clear that there are five NPI scenarios (S1-S5) that you consider, and these scenarios vary in their stringency.
- 7) "leads to modest reductions in R for across "  "leads to modest reductions in R across"
- 8) A general question throughout: how do you derive confidence intervals? Repeatedly rerunning the simulation? This should be explained somewhere.
- 9) Table 2 is a key result, and the main finding is that other than S5, control is very unlikely. Perhaps some color-coding could highlight this finding. And it would be useful to have a reminder in the caption of the key differences between S5, S4, and S3.
- 10) Looking at Figure 3, I wonder if I've missed something. S5 corresponds to full lockdown---but it seems that social distancing does not get $R < 1$ in that future. How could that be? Or should I assume that the purple hatched bar (isolation of index case) implicitly applies during full lockdown? I think that must be it--in which case, it's a substantive finding about what it is that makes a lockdown effective which is worth highlighting, though I'm not entirely sure how to explain it. Conversely, it would be good to have a good explanation for why TTI on its own is nowhere near enough to get $R < 1$.

11) I was confused by the discussion of Figure 4 (Bottom Left). The differences from 3,3 to 1,1 are rather minimal, though I agree with your conclusion that these delays should be reduced.

12) It's important to be clear throughout on what your definition is of "cases." E.g. I think Section G should really be number of infections, not cases. (But as long as you're consistent I'm happy.)

Decision letter (RSOS-201491.R0)

Dear Mr Paleyes

On behalf of the Editors, we are pleased to inform you that your Manuscript RSOS-201491 "Effectiveness and Resource Requirements of Test, Trace and Isolate Strategies" has been accepted for publication in Royal Society Open Science subject to minor revision in accordance with the referees' reports. Please find the referees' comments along with any feedback from the Editors below my signature. Additionally, we sincerely apologise for the delays incurred during peer-review of your manuscript. It was increasingly difficult to secure referees for your paper, and thank you for your continued patience with this process.

Please submit your revised manuscript and required files (see below) no later than 7 days from today's (ie 25-Jan-2021) date. Note: the ScholarOne system will 'lock' if submission of the revision is attempted 7 or more days after the deadline. If you do not think you will be able to meet this deadline please contact the editorial office immediately.

on behalf of Professor Matjaz Perc (Associate Editor) and Mark Chaplain (Subject Editor)
openscience@royalsociety.org

Associate Editor Comments to Author (Professor Matjaz Perc):

This is an interesting and comprehensive work, but the authors should do better in the introduction, which is very modest and could be easily improved without much effort. Many research works have recently looked at the spreading of COVID-19, for example Early spread of COVID-19 in Romania: Imported cases from Italy and human-to-human transmission networks, *R. Soc. Open Sci.* 7, 200780 (2020) and City size and the spreading of COVID-19 in Brazil, *PLoS ONE* 15, e0239699 (2020). It would also improve the paper if the figure captions would be made more self-contained. The title could also more accurately reflect that this is dealing with COVID-19.

Reviewer comments to Author:

Reviewer: 1

Comments to the Author(s)

Abstract:

This is a timely and compelling piece of work. It is well-presented, including both the strengths, and limitations. I have a number of minor comments below, all of which should be straightforward to address.

1) "We further show that adding contact tracing for extra-household contacts of confirmed cases to this broader package" -- clarify what this means

2) " we use contact simulation model " -> " we use *a* contact simulation model "

3) For your discussion, worth citing

<https://www.medrxiv.org/content/10.1101/2020.09.15.20191957v1.full.pdf> in the context of your assumption of 80% compliance. Since it looks pretty easy to rerun your simulations, you could take some of the parameters from that study and do an extra analysis showing that TTI is (presumably) ineffective based on their study.

4) Section 2, paragraph 1, missing reference

5) [10] should be updated to the Nature publication. The 20k number seems perfectly reasonable, but you should clarify exactly where it came from---neither report 13 (currently cited) or Flaxman et al (Nature) extend to 10 May. For context, the most recent ONS report says upper bound 8,300 cases per day in England. Scaling that up to the UK gives about 10k cases per day. (Returning to this comment, you'll still need to cite [10] for the 3.87 R0 you use throughout.)

6) The phrasing of "We consider five scenarios of combinations of NPIs with varying stringency levels." confused me, especially as it appears just below Table 1. Perhaps present TTI strategies first? Either way, rephrase that sentence to make it clear that there are five NPI scenarios (S1-S5) that you consider, and these scenarios vary in their stringency.

7) "leads to modest reductions in R for across "  "leads to modest reductions in R across"

8) A general question throughout: how do you derive confidence intervals? Repeatedly rerunning the simulation? This should be explained somewhere.

9) Table 2 is a key result, and the main finding is that other than S5, control is very unlikely.

Perhaps some color-coding could highlight this finding. And it would be useful to have a reminder in the caption of the key differences between S5, S4, and S3.

10) Looking at Figure 3, I wonder if I've missed something. S5 corresponds to full lockdown--- but it seems that social distancing does not get $R < 1$ in that future. How could that be? Or should I assume that the purple hatched bar (isolation of index case) implicitly applies during full lockdown? I think that must be it--in which case, it's a substantive finding about what it is that makes a lockdown effective which is worth highlighting, though I'm not entirely sure how to explain it. Conversely, it would be good to have a good explanation for why TTI on its own is nowhere near enough to get $R < 1$.

11) I was confused by the discussion of Figure 4 (Bottom Left). The differences from 3,3 to 1,1 are rather minimal, though I agree with your conclusion that these delays should be reduced.

12) It's important to be clear throughout on what your definition is of "cases." E.g. I think Section G should really be number of infections, not cases. (But as long as you're consistent I'm happy.)

===PREPARING YOUR MANUSCRIPT===

===PREPARING YOUR REVISION IN SCHOLARONE===

To revise your manuscript, log into <https://mc.manuscriptcentral.com/rsos> and enter your Author Centre - this may be accessed by clicking on "Author" in the dark toolbar at the top of the

page (just below the journal name). You will find your manuscript listed under "Manuscripts with Decisions". Under "Actions", click on "Create a Revision".

<https://royalsociety.org/journals/authors/author-guidelines/#supplementary-material> to include a suitable title and informative caption. An example of appropriate titling and captioning may be found at https://figshare.com/articles/Table_S2_from_Is_there_a_trade-off_between_peak_performance_and_performance_breadth_across_temperatures_for_aerobic_sc_ope_in_teleost_fishes_/3843624.

Author's Response to Decision Letter for (RSOS-201491.R0)

See Appendix A.

Decision letter (RSOS-201491.R1)

Dear Mr Paleyes,

It is a pleasure to accept your manuscript entitled "Effectiveness and Resource Requirements of Test, Trace and Isolate Strategies for COVID in the UK" in its current form for publication in Royal Society Open Science. The comments of the reviewer(s) who reviewed your manuscript are included at the foot of this letter.

COVID-19 rapid publication process:

We are taking steps to expedite the publication of research relevant to the pandemic. If you wish, you can opt to have your paper published as soon as it is ready, rather than waiting for it to be published the scheduled Wednesday.

This means your paper will not be included in the weekly media round-up which the Society sends to journalists ahead of publication. However, it will still appear in the COVID-19 Publishing Collection which journalists will be directed to each week (<https://royalsocietypublishing.org/topic/special-collections/novel-coronavirus-outbreak>).

If you wish to have your paper considered for immediate publication, or to discuss further, please notify openscience_proofs@royalsociety.org and press@royalsociety.org when you respond to this email.

Please see the Royal Society Publishing guidance on how you may share your accepted author manuscript at <https://royalsociety.org/journals/ethics-policies/media-embargo/>. After publication, some additional ways to effectively promote your article can also be found here

<https://royalsociety.org/blog/2020/07/promoting-your-latest-paper-and-tracking-your-results/>.

on behalf of Professor Matjaz Perc (Associate Editor) and Mark Chaplain (Subject Editor)
openscience@royalsociety.org

Associate Editor Comments to Author (Professor Matjaz Perc):

Comments to the Author:

Thank you for submitting your revised manuscript, which we are happy to accept for publication in Royal Society Open Science.

Appendix A

Editor comments

1. The authors should do better in the introduction, which is very modest and could be easily improved without much effort.

The intro is completely rewritten.

2. Many research works have recently looked at the spreading of COVID-19, for example Early spread of COVID-19 in Romania: Imported cases from Italy and human-to-human transmission networks, R. Soc. Open Sci. 7, 200780 (2020) and City size and the spreading of COVID-19 in Brazil, PLoS ONE 15, e0239699 (2020).

We have referred to both of these works in our related work.

3. It would also improve the paper if the figure captions would be made more self-contained.

We improved captions of several figures.

4. The title could also more accurately reflect that this is dealing with COVID-19.

“for COVID in the UK” appended to title.

Reviewer comments

1. "We further show that adding contact tracing for extra-household contacts of confirmed cases to this broader package" -- clarify what this means

Changed sentence to “ We further show that adding contact tracing of non-household contacts of confirmed cases to this broader package of interventions reduces the number of new infections otherwise generated by 5-15%.”

2. " we use contact simulation model " -> " we use *a* contact simulation model "

Added “a”

3. For your discussion, worth citing <https://www.medrxiv.org/content/10.1101/2020.09.15.20191957v1.full.pdf> in the context of your assumption of 80% compliance. Since it looks pretty easy to rerun your simulations, you could take some of the parameters from that study and do an extra analysis showing that TTI is (presumably) ineffective based on their study.

Added citation, and description of how it complements our findings on the importance of maximising compliance for TTI effectiveness.

Added references to figures 4 and 10 to our sensitivity plots for compliance.

4. Section 2, paragraph 1, missing reference

Reference added

- [10] should be updated to the Nature publication. The 20k number seems perfectly reasonable, but you should clarify exactly where it came from---neither report 13 (currently cited) or Flaxman et al (Nature) extend to 10 May. For context, the most recent ONS report says upper bound 8,300 cases per day in England. Scaling that up to the UK gives about 10k cases per day. (Returning to this comment, you'll still need to cite [10] for the 3.87 R_0 you use throughout.)

Updated [10] to Nature publication, and also changed date for 20k number to 4 May which is reported in [10].

- The phrasing of "We consider five scenarios of combinations of NPIs with varying stringency levels." confused me, especially as it appears just below Table 1. Perhaps present TTI strategies first? Either way, rephrase that sentence to make it clear that there are five NPI scenarios (S1-S5) that you consider, and these scenarios vary in their stringency.

Table moved to bottom of page to remove confusion from this.

Sentenced change to "We consider five scenarios, each with a different combination of NPIs, corresponding to varying levels of stringency."

- "leads to modest reductions in R for across "  "leads to modest reductions in R across"

fixed

- A general question throughout: how do you derive confidence intervals? Repeatedly rerunning the simulation? This should be explained somewhere.

Paragraph added in Section D

- Table 2 is a key result, and the main finding is that other than S5, control is very unlikely. Perhaps some color-coding could highlight this finding. And it would be useful to have a reminder in the caption of the key differences between S5, S4, and S3.

We have emphasized $R < 1$ with bold font, and added some clarification text to the table caption.

- Looking at Figure 3, I wonder if I've missed something. S5 corresponds to full lockdown-- but it seems that social distancing does not get $R < 1$ in that future. How could that be? Or should I assume that the purple hatched bar (isolation of index case) implicitly applies during full lockdown? I think that must be it--in which case, it's a substantive

finding about what it is that makes a lockdown effective which is worth highlighting, though I'm not entirely sure how to explain it. Conversely, it would be good to have a good explanation for why TTI on its own is nowhere near enough to get $R < 1$.

“Hatched bars indicate infections \textit{prevented} by each stringency level” added to fig 3 caption

Clarifying statement “Note we have separated the effects of base social distancing (reducing social contacts, closure of shops etc.) from isolating index cases to demonstrate the scenario in which regular social distancing is not used, but index cases are still isolated e.g. S5. The reversed scenario is unlikely to ever be used. The combination of these two is what would colloquially be referred to as "social distancing"” added

11. I was confused by the discussion of Figure 4 (Bottom Left). The differences from 3,3 to 1,1 are rather minimal, though I agree with your conclusion that these delays should be reduced.

Extra discussion added.

12. It's important to be clear throughout what your definition is of "cases." E.g. I think Section G should really be the number of infections, not cases. (But as long as you're consistent I'm happy.)

Section G is using “infections” now. We also reviewed our use of “cases” in other place throughout the paper